# Structural insights into the complex of oncogenic KRas4B^G12V and Rgl2, a RalA/B activator

Mishal Tariq[1], Teppei Ikeya[2], Naoyuki Togashi[2], Louise Fairall[1,3], Shun Kamei[2], Sannojah Mayooramurugan[1], Lauren R Abbott[1], Anab Hasan[1], Carlos Bueno-Alejo[3], Sakura Sukegawa[2], Beatriz Romartinez-Alonso[1,3], Miguel Angel Muro Campillo[1], Andrew J Hudson[3,4], Yutaka Ito[2], John WR Schwabe[1,3], Cyril Dominguez[1,3], Kayoko Tanaka[1]

**About a quarter of total human cancers carry mutations in Ras isoforms. Accumulating evidence suggests that small GTPases, RalA, and RalB, and their activators, Ral guanine nucleotide exchange factors (RalGEFs), play an essential role in oncogenic Ras-induced signalling. We studied the interaction between human KRas4B and the Ras association (RA) domain of Rgl2 (Rgl2^RA), one of the RA-containing RalGEFs. We show that the G12V oncogenic KRas4B mutation changes the interaction kinetics with Rgl2^RA. The crystal structure of the KRas4B^G12V: Rgl2^RA complex shows a 2:2 heterotetramer where the switch I and switch II regions of each KRas^G12V interact with both Rgl2^RA molecules. This structural arrangement is highly similar to the HRas^E31K:RALGDS^RA crystal structure and is distinct from the well-characterised Ras: Raf complex. Interestingly, the G12V mutation was found at the dimer interface of KRas4B^G12V with its partner. Our study reveals a potentially distinct mode of Ras:effector complex formation by RalGEFs and offers a possible mechanistic explanation for how the oncogenic KRas4B^G12V hyperactivates the RalA/B pathway.**

## Introduction

Ras belongs to a family of small G protein that switches between two states, GDP-bound form (Ras^GDP) and GTP-bound form (Ras^GTP), and regulates a wide range of cellular activities (Cox & Der, 2010). Guanine nucleotide exchange factors (GEFs), typically activated by growth factor signalling, mediate the conversion of Ras^GDP to Ras^GTP, whereas the GTP-bound status lasts only transiently as the intrinsic GTPase activity, aided by the GTPase-activating proteins, hydrolyses the bound GTP into GDP (Vetter & Wittinghofer, 2001). Ras acts as a signalling hub where the Ras^GTP, but not Ras^GDP, physically interacts with multiple Ras effectors, which then transmits the signal to downstream molecules, including ERK, Akt, and small G proteins RalA and RalB (Simanshu et al, 2017).

There are three human *RAS* genes, *KRAS*, *NRAS*, and *HRAS*, and as *KRAS* is alternatively spliced at exon 4, the three *RAS* genes produce four Ras isoforms KRas4A, KRas4B, NRas, and HRas (Castellano & Santos, 2011). Among them, *KRAS4B* typically represents more than half of all *RAS* transcripts (Newlaczyl et al, 2017). The Catalogue Of Somatic Mutations In Cancer (COSMIC) shows that about 25% of all cancers carry mutations in *RAS* genes, and *KRAS* is responsible for about 70% of these mutations (COSMIC, v.95). Hence, it is vital to obtain more insights into how the oncogenic KRas signalling leads to cancers.

Extensive earlier biochemical studies revealed that oncogenic Ras mutations cause a reduction of the GTP hydrolysis rate, generating an increased population of Ras^GTP (Gibbs et al, 1984; McGrath et al, 1984; Manne et al, 1985; Der et al, 1986; Bollag & McCormick, 1991; Scheffzek et al, 1997). This likely leads to an overactivation of the downstream signalling pathways and therefore is considered the major cause of the KRas oncogenicity. In addition, recent biochemical, structural, and molecular modelling studies indicate that oncogenic Ras mutations affect the interaction kinetics with the effector molecules and may trigger a biased overactivation of a set of effectors (Smith & Ikura, 2014; Hunter et al, 2015; Mazhab-Jafari et al, 2015; Pantsar et al, 2018). Therefore, obtaining more insights into the Ras-effector interaction mechanisms are essential to understand the oncogenic Ras–mediated tumorigenesis process.

Effector molecules that have been shown to interact with Ras all carry a domain consisting of an ubiquitin-fold structure, although the primary sequences of these domains show a limited homology (Kiel & Serrano, 2006). They are classified into three categories according to Uniprot PROSITE-ProRule annotation based on their primary sequences; Ras-associating (RA) (PRU00166), Ras-binding domain (RBD) (PRU002262), and phosphatidylinositol 3-kinase Ras-binding domain (PI3K-RBD) (PRU00879). In this study, we follow these annotations.

[1]Department of Molecular and Cell Biology, University of Leicester, Leicester, UK   [2]Department of Chemistry, Tokyo Metropolitan University, Hachioji, Japan   [3]Leicester Institute of Structure and Chemical Biology, University of Leicester, Leicester, UK   [4]Department of Chemistry, University of Leicester, Leicester, UK

Correspondence: kt96@le.ac.uk

Among Ras effectors, Raf kinase and PI3K, which triggers ERK and Akt signalling, respectively, have attracted much attention, especially because they were found not only to be activated by oncogenic Ras but also to be able to carry oncogenic mutations themselves (Chalhoub & Baker, 2009; Maurer et al, 2011). However, accumulating evidence suggests that the misregulation of Ral GTPases, RalA, and RalB, rather than ERK and Akt signalling pathways, may act as the initial trigger of oncogenic Ras–induced tumorigenesis. For example, in the oncogenic K-ras(G12D) knock-in mouse model, although neoplastic and developmental defects were observed, hyper-activation of ERK or Akt was undetected (Tuveson et al, 2004). In humans, ERK hyper-activation is often missing in cancer cell lines and tissues with oncogenic *RAS* mutations, whereas RalA and RalB are essential in oncogenic Ras–induced cell proliferation, motility, and tumorigenesis (Miller et al, 1997; Luo & Sharif, 1999; Yip-Schneider et al, 1999, 2001; Lim et al, 2005, 2006; Campbell et al, 2007; Zago et al, 2018). Therefore, a biased misregulation of the Ral GTPases may be the critical feature behind Ras oncogenicity. Indeed, nuclear magnetic resonance (NMR)-based effector competition assays suggested the intriguing possibility that the oncogenic Ras molecules may develop an altered effector preference, leading to a biased hyperactivation of RalA (Smith & Ikura, 2014).

Activation of RalA and RalB is mediated by guanine nucleotide exchange factors (RalGEFs). There are eight RalGEFs reported for humans, and four of them, RALGDS, Rgl1, Rgl2, and Rgl3, have a Ras-association (RA) domain responsible for Ras-binding (Apken & Oeckinghaus, 2021). RALGDS was the first to be identified as a Ras effector among these RalGEFs and is by far the most studied (Neel et al, 2011). RALGDS is one of the first effector molecules that was crystallised together with an active Ras; rat RALGDS was co-crystalised with human HRas harbouring an E31K mutation, which helped complex formation (Huang et al, 1998). Meanwhile, a recent modelling approach, integrating proteomic data of Ras, its 56 effectors and their interaction affinities, predicts that Rgl2 would generate the highest concentration of Ras:effector complex among RalGEFs in 28 out of 29 healthy human tissues (Catozzi et al, 2021). Furthermore, Rgl2 plays a critical role in oncogenic Ras–induced tumour phenotypes (Vigil et al, 2010), and overexpression of the Rgl2^RA interferes with Ras-mediated signaling, likely by competitively titrating active Ras molecules (Peterson et al, 1996; Fischer et al, 2003).

Human Rgl2 and its mouse homologue, Rlf, were initially identified through yeast two-hybrid screenings and were shown to interact with Ras through the RA domain in vitro (Esser et al, 1998; Ferro et al, 2008; O'Gara M et al, 1997; Peterson et al, 1996; Wolthuis et al, 1996). Furthermore, in vivo interaction of Rgl2 and Ras was reported in a BioID-based proteomics study where Rgl2 was among other proximal interactors identified using HRas.G12V expressed in bladder cancer cells (Kovalski et al, 2019). In addition, the full-length Rlf was shown to interact with HRas.G12V and acted as a RalGEF when expressed in COS-7 cells (Wolthuis et al, 1997). However, the Ras:Rgl2 complex interface at an atomic level has not been explored. Furthermore, whether oncogenic mutations influence the interaction kinetics or not is unclear, making it difficult to appreciate the impact of an oncogenic mutation in activating the

RalGEF signalling branch that plays an essential role in the oncogenic Ras-mediated tumorigenesis.

To address the question, we examined the mode of interaction between human KRas4B and the RA of Rgl2 (Rgl2^RA) in this report. We observed a change in the interaction kinetics between KRas4B and the Rgl2^RA upon the introduction of the G12V oncogenic mutation. Our crystal structure of KRas4B^G12V:Rgl2^RA complex shows a heterotetramer formation, highly similar to the reported HRas^E31K:RALGDS^RA complex, but distinct from other Ras:effector complexes, including Ras:Raf1. Interestingly, in our structure, the G12V oncogenic mutation is located at the dimer interface of KRas4B^G12V with its homodimeric partner. Our findings provide an interesting possibility that KRas4B^G12V oncogenicity might be contributed by the altered interaction kinetics with Rgl2.

## Results

### The G12V oncogenic mutation causes a change in the interaction kinetics between the KRas4B and the Rgl2^RA

Throughout this study, we used recombinant bacteria constructs of human KRas4B lacking the C-terminal hyper-variable region and the Rgl2^RA consisting of amino acids 643–740 of human Rgl2 (Fig S1A). The binding of Rgl2^RA and KRas4B^G12V was confirmed for the active KRas4B^G12V loaded with a non-hydrolysable GTP analogue (Guanosine-5'-[(β,γ)-imido]triphosphate [GMPPNP]), but not with the GDP-loaded KRas4B^G12V, by GST-pulldown assays (Fig S1B). To examine whether the binding mode of Rgl2^RA differs between KRas4B^WT and KRas4B^G12V, the binding kinetics were measured by biolayer interferometry (BLI). We noticed that both KRas4B^WT and KRas4B^G12V, purified at 4°C at all times, retained GTP as the major bound guanine nucleotide. In addition, in our hands, the efficacy of GMPPNP loading could vary between samples, whereas the efficacy of GTP loading was highly reproducible. We also confirmed that the KRas4B^WT and KRas4B^G12V samples that were subjected to the incubation condition of the BLI assay (20°C for 30 min) were still associated with GTP (Fig 1A). Therefore, we loaded the samples with GTP and used these GTP-bound samples for the BLI experiment so that we could minimize possible artefacts caused by GMPPNP.

The sensorgram curves representing the on/off kinetics between KRas4B and Rgl2^RA were distinct when using the KRas4B^WT or KRas4B^G12V proteins (Fig 1B). When the KRas4B–Rgl2^RA-binding kinetics results were model-fitted using a 1:1 binding model, the *kon* and *koff* values were larger in the case of the G12V mutant, indicating more dynamic interaction between the G12V mutant and Rgl2^RA (Fig 1B). Meanwhile, the K_D values were comparable between the KRas4B^WT (about 1.48 μM) and KRas4B^G12V (about 1.49 μM) (Fig 1B), and the elution profiles of the size exclusion chromatography (SEC) showed little difference between the wildtype and the G12V mutant (Fig S1D and E).

Interestingly, the KRas4B^WT–Rgl2^RA sensorgram curve was more consistent with a 2:1 heterogeneous binding model fitting or a 1:2 bivalent binding model fitting (Data Analysis HT Software, Sartorius), as indicated by the improvement of the residual sum of squares (RSS) value; from about 90.5 (the 1:1 model, Fig 2B) to about

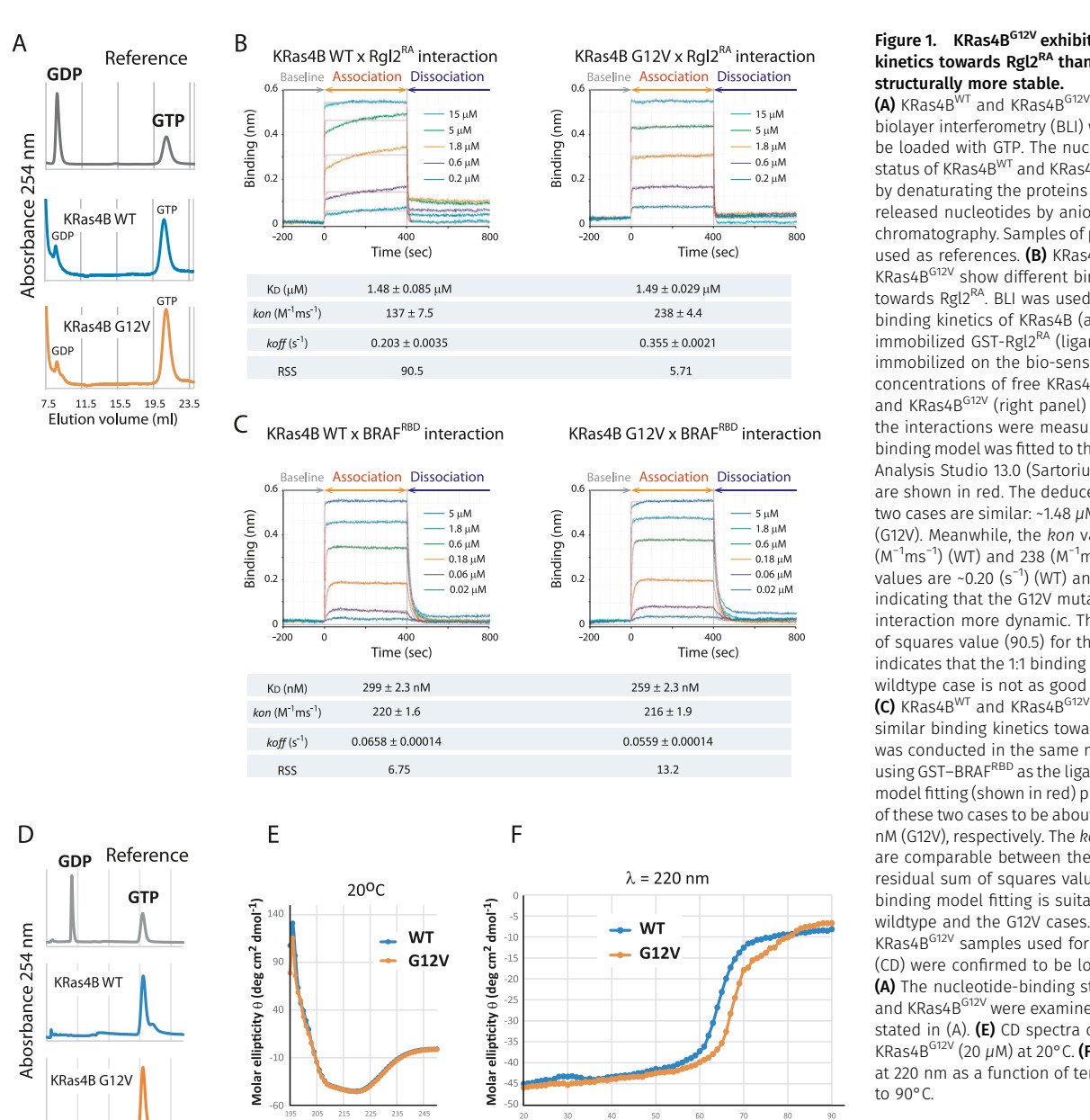

**Figure 1. KRas4B^G12V exhibits a different binding kinetics towards Rgl2^RA than KRas4B^WT and is structurally more stable.**
**(A)** KRas4B^WT and KRas4B^G12V samples used for biolayer interferometry (BLI) were confirmed to be loaded with GTP. The nucleotide-binding status of KRas4B^WT and KRas4B^G12V were examined by denaturing the proteins and detecting the released nucleotides by anion exchange chromatography. Samples of pure GDP or GTP were used as references. **(B)** KRas4B^WT and KRas4B^G12V show different binding kinetics towards Rgl2^RA. BLI was used to measure the binding kinetics of KRas4B (analyte) to immobilized GST-Rgl2^RA (ligand). GST-Rgl2^RA was immobilized on the bio-sensors, and varying concentrations of free KRas4B^WT (left panel) and KRas4B^G12V (right panel) were provided, and the interactions were measured at 20°C. The 1:1 binding model was fitted to the data using Octet Analysis Studio 13.0 (Sartorius). The fitted curves are shown in red. The deduced $K_D$ values of these two cases are similar: ~1.48 $\mu$M (WT) and 1.49 $\mu$M (G12V). Meanwhile, the $kon$ values are ~137 (M$^{-1}$ms$^{-1}$) (WT) and 238 (M$^{-1}$ms$^{-1}$) (G12V), and $koff$ values are ~0.20 (s$^{-1}$) (WT) and 0.36 (s$^{-1}$) (G12V), indicating that the G12V mutation may cause the interaction more dynamic. The high residual sum of squares value (90.5) for the wildtype case indicates that the 1:1 binding model fitting for the wildtype case is not as good as for the G12V case. **(C)** KRas4B^WT and KRas4B^G12V show highly similar binding kinetics towards BRAF^RBD. **(B)** BLI was conducted in the same manner as (B), except using GST–BRAF^RBD as the ligand. The 1:1 binding model fitting (shown in red) predicts the $K_D$ values of these two cases to be about 299 nM (WT) and 259 nM (G12V), respectively. The $kon$ and $koff$ values are comparable between the two cases, and the residual sum of squares values ensure the 1:1 binding model fitting is suitable for both the wildtype and the G12V cases. **(D)** KRas4B^WT and KRas4B^G12V samples used for circular dichroism (CD) were confirmed to be loaded with GTP. **(A)** The nucleotide-binding status of KRas4B^WT and KRas4B^G12V were examined in the same way as stated in (A). **(E)** CD spectra of KRas4B^WT and KRas4B^G12V (20 $\mu$M) at 20°C. **(F)** CD signal intensity at 220 nm as a function of temperature from 20°C to 90°C.

1.05 (the 2:1 model, Fig S2A) or 1.85 (the 1:2 model, Fig S2A). Therefore, the KRas4B^WT–Rgl2^RA interaction does not correspond to a simple 1:1 binding mode. In contrast, for the KRas4B^G12V–Rgl2^RA sensorgram curve, the RSS values for the 1:1 and the 2:1 model fittings were comparable; about 5.71 for the 1:1 model (Fig 1B) and about 4.40 for the 2:1 model (Fig S2B). Furthermore, in the 2:1 heterogeneous model, "Binding type 2" represents 95–100% of the whole population, where the values for the $K_D$, $kon$, and $koff$ are all similar to the ones deduced from the 1:1 binding model (Fig S2B). These results strongly indicate that the binding of the KRas4B^G12V–Rgl2^RA complex is compatible with a 1:1 binding.

The difference in kinetics of binding observed for KRas4B^WT and KRas4B^G12V mutant with Rgl2^RA contrasts with the on/off kinetics for

KRas4B^WT and KRas4B^G12V mutant with BRAF^RBD, a well-established Ras effector. In that case, BRAF^RBD interacted with both the wildtype and the G12V mutant in a similar manner (Fig 1C). The fitting using a 1:1 binding model was adequate for both KRas4B^WT and KRas4B^G12V cases (the RSS values are about 6.75 and 13.2, respectively, Fig 1C) and the 2:1 binding model only marginally improved the fitting (the RSS values are about 2.96 and 4.34, respectively, Fig S2C).

These results indicate that the G12V oncogenic mutation may impact more significantly on the Rgl2-mediated signalling pathway and highlight the distinct binding mode of KRas4B^WT and KRas4B^G12V mutant to Rgl2^RA compared with BRAF^RBD.

The circular dichroic (CD) spectra for KRas4B^WT and KRas4B^G12V samples across the increasing temperature from 20°C to 90°C

**Figure 2.  Crystal structure of the KRas4B^G12V:Rgl2^RA 2:2 heterotetramer.**
Cartoon representation of the structure of the heterotetramer complex of KRas4B^G12V and Rgl2^RA with top view and side view; the two KRas4B^G12V molecules are shown in dark and pale cyan and the two molecules of Rgl2^RA in pink and violet. Switch I and switch II regions of KRas4B^G12V are shown in green and yellow, respectively, and the α-helix and β-sheets are numbered for each chain. The Mg^2+ is shown as a grey sphere. The structure shows that each KRas4B^G12V molecule interacts with two Rgl2^RA molecules (referred to as I and II) at switch I and switch II individually.

showed improved structural stability for KRas4B^G12V, which might contribute to the altered binding kinetics towards Rgl2^RA (Fig 1D–F).

## A 2:2 tetramer of KRas4B^G12V:Rgl2^RA complex in the crystal structure

We conducted crystallization trials to obtain structural insights into the KRas4B:Rgl2^RA complex. To purify the complex, a mixture of GMPPNP-loaded KRas4B and Rgl2^RA was applied onto a size-exclusion column. The major elution peak corresponded to the co-elution of KRas4B^G12V and Rgl2^RA as a complex, and, as expected, the complex eluted earlier than KRas4B^G12V or Rgl2^RA alone (Fig S1C).

Crystals of the KRas4B^G12V:Rgl2^RA complex were obtained which diffracted at a resolution of 3.07 Å (Table 1). The phase was solved by molecular replacement using the structure of the human HRas^E31K: rat RALGDS^RA complex (PDB ID 1LFD) (Huang et al, 1998). The space group was assigned to P12₁1 with four proteins per asymmetric unit (two molecules of KRas^G12V and two molecules of Rgl2^RA) arranged as a tetramer (Fig 2). The overall arrangement of KRas4B^G12V:Rgl2^RA crystal structure is similar to HRas^E31K:RALGDS^RA crystal structure (PDB ID 1LFD [Huang et al, 1998]) (Fig S3A–C). The complex forms a 2:2 heterotetramer where β2 (within switch I) of KRas^G12V and β2 of Rgl2^RA generate a continuously extended β-sheet, along with interaction at switch II of the same KRas^G12V molecule with the second Rgl2^RA molecule (Figs 3A–C and 4A and B). In both KRas4B^G12V:Rgl2^RA and HRas^E31K:RALGDS^RA cases, the Ras–Ras interface is formed within the region spanning amino acids 1–90 where KRas4B^G12V and HRas^E31K share 100% amino acid sequence identity except for the point mutations G12V and E31K. The KRas4B^G12V:Rgl2^RA heterotetramer complex is stabilized by a network of hydrogen bonds and hydrophobic interactions (summarized in Fig S4A), leading to complimentary surface charges (Fig S4B).

## KRas^G12V:Rgl2^RA interface

KRas4B^G12V residues within the switch I interact with residues in β1, β2, and α1 of Rgl2^RA. These interactions include a salt bridge between E37 of KRas4B^G12V switch I and R653 of Rgl2^RA (Fig 3A and B). The switch II of the same KRas4B^G12V molecule also contributes to the complex formation by interacting with the second Rgl2^RA through residues in β2 and α1 (Fig 3A and C).

## KRas^G12V: KRas^G12V interface

Meanwhile, two KRas4B^G12V molecules have direct contact through switch I, switch II, and α3 (Figs 4A and S4A). Importantly, the V12 residue of the oncogenic G12V mutation is in the proximity of the ring of Y32 of the neighbouring KRas4B^G12V, contributing to the KRas4B^G12V: KRas4B^G12V interface, increasing the hydrophobic pocket.

## Rgl2^RA:Rgl2^RA interface

The β1 of both Rgl2^RA molecules run anti-parallel to each other, interacting through various hydrophobic interactions and hydrogen bonds at both side-chain and backbone levels (Figs 4B and S4A).

## Solution NMR analyses of the KRas4B^G12V:Rgl2^RA complex

The KRas4B^G12V:Rgl2^RA complex was furthermore analyzed in solution by NMR. First, the solution structure of free Rgl2^RA was determined by solution NMR (Fig S5A) (Table 2). As expected, the structure adopts the ββαββαβ ubiquitin-fold structure, a common feature for the RA/RBDs (Kiel & Serrano, 2006). Overall, it is similar to the structure of Rlf, the mouse homologue of human Rgl2 (PDB ID 1RLF) (Esser et al, 1998) (Fig S5B). The solution structure of free Rgl2^RA is also very similar to the crystal structure of Rgl2^RA in the KRas4B^G12V:Rgl2^RA, indicating that the complex formation causes relatively small structural changes, and the crystal structure likely reflects the physiological Rgl2^RA folding (Fig 5A).

The KRas4B^G12V:Rgl2^RA complex was next analyzed in solution by NMR chemical shift perturbation. Two-dimensional (2D) ^15N–^1H-heteronuclear single quantum coherence (HSQC) spectra were measured for the ^15N-labelled Rgl2^RA sample in the absence or presence of an increasing amount of non-labelled KRas4B^G12V (Fig 5B and C). Chemical shift perturbations were observed for many of the Rgl2^RA residues, in agreement with KRas4B^G12V:Rgl2^RA complex formation in solution. Most of the Rgl2^RA residues at the KRas4B^G12V: Rgl2^RA and Rgl2^RA:Rgl2^RA interfaces in the crystal structure showed greater changes (either decreased or increased) in their NMR signal intensities and in chemical shift perturbation, indicating their participation in the complex formation and supporting the heterodimer interface observed in the crystal structure (Fig 5B and C).

**Table 1.** Data collection and refinement statistics for human KRas4B^(G12V): Rgl2^(RA) complex.

| Wavelength | 0.9795 |
|---|---|
| Resolution range | 28.68–3.071 (3.18–3.071) |
| Space group | P 32 2 1 |
| Unit cell | 77.843 77.843 163.728 90 90 120 |
| Total reflections | 106,507 (7,598) |
| Unique reflections | 11,107 (949) |
| Multiplicity | 9.6 (8.0) |
| Completeness (%) | 98.22 (86.51) |
| Mean I/sigma(I) | 11.15 (1.64) |
| Wilson B-factor | 95.59 |
| R-merge | 0.1313 (0.6732) |
| R-meas | 0.1389 (0.7159) |
| R-pim | 0.04474 (0.2379) |
| CC1/2 | 0.998 (0.957) |
| CC* | 1 (0.989) |
| Reflections used in refinement | 11,065 (936) |
| Reflections used for R-free | 582 (41) |
| R-work | 0.2367 (0.3229) |
| R-free | 0.3204 (0.3572) |
| CC (work) | 0.943 (0.897) |
| CC (free) | 0.851 (0.486) |
| Number of non-hydrogen atoms | 4,116 |
| Macromolecules | 4,050 |
| Ligands | 66 |
| Solvent | 0 |
| Protein residues | 514 |
| RMS (bonds) | 0.007 |
| RMS (angles) | 1.40 |
| Ramachandran favored (%) | 95.22 |
| Ramachandran allowed (%) | 3.98 |
| Ramachandran outliers (%) | 0.80 |
| Rotamer outliers (%) | 0.89 |
| Clashscore | 14.47 |
| Average B-factor | 113.50 |
| Macromolecules | 113.74 |
| Ligands | 98.55 |

Statistics for the highest-resolution shell are shown in parentheses.

Signals from the Rgl2^(RA) residues in β1 at the N-terminal end (642–649) display the largest changes. This is in agreement with the KRas4B^(G12V):Rgl2^(RA) heterotetramer crystal structure where the highly flexible N-terminal region of free Rgl2^(RA) (Fig S5A) becomes rigid through interaction with another Rgl2 molecule in the hetero-tetramer (Fig 4B). In addition, the overall decrease in the NMR signals is compatible with the formation of a relatively large

complex such as the KRas4B^(G12V):Rgl2^(RA) heterotetramer complex (62 kD). Meanwhile, the titration experiment showed that the chemical shift changes were not fully saturated in most of the Rgl2^(RA) residues when the Rgl2^(RA):Kras^(G12V) molar ratio was 1:1 or even 1:2 (Fig 5C), suggesting that the stoichiometry is more complex than simple 1:1, possibly due to the heterotetramer formation.

We next examined the NMR chemical shift perturbation of KRas4B^(G12V) in solution in the absence or presence of non-labelled Rgl2^(RA) (Fig 6). When loaded with GMPPNP, the number of detectable peaks of the $^{15}$N-labelled KRas4B^(G12V) signal is substantially decreased compared with the number of signals obtained from the GDP-loaded sample (Fig S6) as previous studies reported (Ito et al, 1997; Menyhard et al, 2020). Consequently, unfortunately, most of the KRas4B^(G12V) residues at the KRas4B^(G12V):Rgl2^(RA) complex interface could not be detected (Fig 6A and B), except for K42 and K88, both of which showed substantially decreased signal intensities upon Rgl2^(RA) addition, in agreement with their expected participation in the complex formation (Fig 6C). Furthermore, the addition of Rgl2^(RA) leads to a decrease of most of the KRas4B^(G12V) signals further suggesting formation of a relatively large complex, like the heterotetramer.

To estimate the size of the complex, we conducted NMR relaxation measurements. The $^{15}$N longitudinal ($T_1$) and transverse ($T_2$) relaxation times, and the steady-state heteronuclear {$^1$H}-$^{15}$N NOE of $^2$H/$^{13}$C/$^{15}$N- labelled KRas4B^(G12V), mixed with non-labelled Rgl2^(RA), was measured with TROSY-based pulse schemes (Fig S7A) (Lakomek et al, 2012). The overall rotational correlation time $\tau_c$ deduced from the measured $T1$, $T2$, and NOE was ~14.7 nsec. Meanwhile, the $\tau_c$ values estimated theoretically for the dimer and the tetramer were ~14.2 and 38.8 nsec when we assumed that they were spherical and the radii of gyration were 25 and 35 Å, respectively (Fig S7B). These results suggest that the status of the KRas4B^(G12V)–Rgl2^(RA) complex in solution may be closer to the heterodimer structure. However, because of the crystal structures are ellipsoidal rather than spherical, and an equilibrium-like exchange process between the heterodimer and heterotetramer, or transient tetramer formation, may occur, furthermore NMR relaxation measurements and detailed analysis would be needed to more accurately predict the complex status.

## Mass photometry indicates the presence of a heterotetramer of KRas4B^(G12V):Rgl2^(RA) complex

To address the question of whether KRas4B^(G12V):Rgl2^(RA) complex can exist as a heterotetramer in solution, we analyzed the complex by mass photometry, a label-free technique that has recently been adapted to measure the mass of biomolecules in solution (Young et al, 2018). Our commercially available instrument (see the Materials and Methods section) can measure masses in the range 30–5,000 kD and is suitable to detect KRas4B^(G12V):Rgl2^(RA) hetero-tetramer (62 kD). The KRas4B^(G12V):Rgl2^(RA) complex was freshly prepared by SEC at 4°C, and the peak fraction that contained both proteins at 1:1 ratio was used for the measurement (Fig 7A). The histogram of the frequency counts for this sample showed a peak corresponding to a biomolecular complex with a molecular weight of 68 kD (Fig 7B). As the expected molecular weight of the

## A    KRas4B<sup>G12V</sup>:Rgl2<sup>RA</sup> interaction

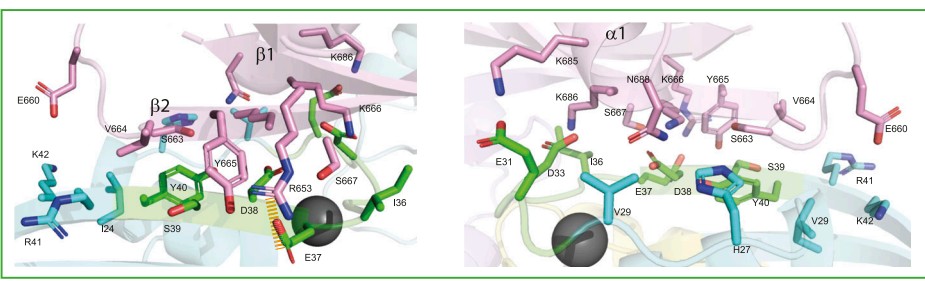

## B    Switch I interaction

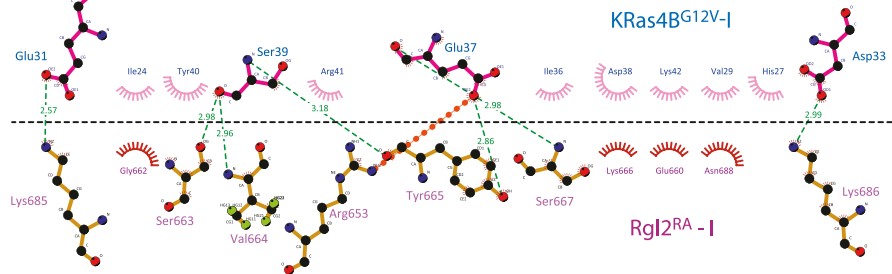

## C    Switch II interaction

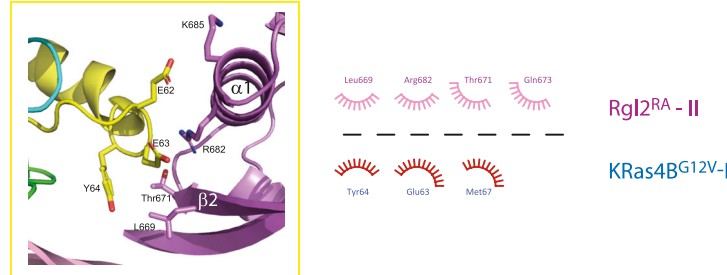

**Figure 3.  The interacting interface of KRas4B$^{G12V}$ and Rgl2$^{RA}$ of the KRas4B$^{G12V}$:Rgl2$^{RA}$ 2:2 heterotetramer.**
**(A)** The overview of the interacting interface of KRas4B$^{G12V}$ and Rgl2$^{RA}$, highlighting the residues involved in hydrogen and hydrophobic interactions between β1, β2, and α1 of Rgl2$^{RA1}$ (violet sticks) and KRas4B$^{G12V}$ switch I (green sticks, enlarged in the green box) or switch II (yellow sticks, enlarged in the yellow box). An orange dashed line shows a salt bridge formed between E37 of KRas4B$^{G12V}$ and R653 of Rgl2$^{RA}$. **(B)** Upper panel: blow-up images of the KRas4B$^{G12V}$: Rgl2$^{RA}$ interface involving switch I. Lower panel: a schematic representation of the intermolecular contacts where hydrogen bonds (green dashed lines), a salt bridge (orange dashed line), and hydrophobic contacts (spiked arches) were predicted by LIGPLOT (Wallace et al, 1995) (lower panel). Numbers indicate atomic distances in Å. **(C)** A blow-up image of the KRas4B$^{G12V}$:Rgl2$^{RA}$ interface involving switch II (left panel) and its LIGPLOT representation (right panel).

KRas4B$^{G12V}$:Rgl2$^{RA}$ heterotetramer is about 62 KD, the result indicates that at least part of the complex population likely exists as the 2:2 heterotetramer in solution.

To detect the presence of the heterodimer and the heterotetramer, we examined the complex formation between KRas4B$^{G12V}$ and a Halo-tagged Rgl2$^{RA}$ (Rgl2$^{RA}$-Halo), where the Halo-tag increased the MW by 35 kD; hence, the heterodimer of KRas4B$^{G12V}$ and

Rgl2$^{RA}$–Halo (expected to be about 66 kD) became within the detection range of the mass photometry instrument. The KRas4B$^{G12V}$ and Rgl2$^{RA}$–Halo were separately prepared and mixed at a 1:1 M ratio at 2 μM, and the 2 μM premix sample was further diluted to 20, 50, 100, and 250 nM for the measurement (Fig 7C). At 20 nM, the MW was about 50 kD, in a similar range to the Rgl2$^{RA}$–Halo monomer. At 50 nM, the MW increased to about 90 kD, which may represent the

## A  KRas4B$^{G12V}$:KRas4B$^{G12V}$ interaction

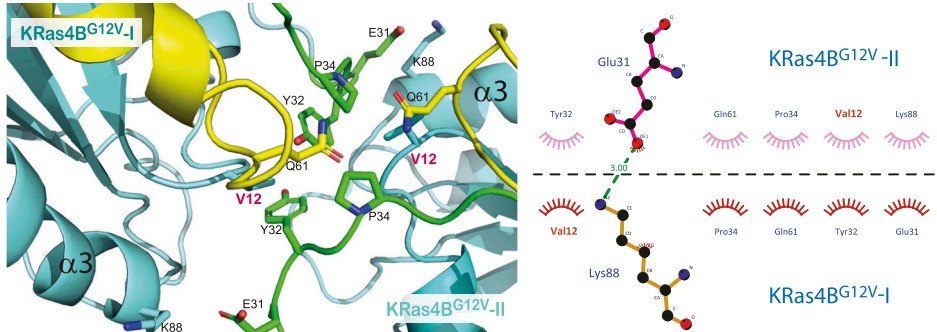

## B  Rgl2$^{RA}$:Rgl2$^{RA}$ interaction

**Figure 4.  The interacting interface of KRas4B$^{G12V}$:KRas4B$^{G12V}$ and Rgl2$^{RA}$:Rgl2$^{RA}$ of the KRas4B$^{G12V}$:Rgl2$^{RA}$ 2:2 heterotetramer. (A)** The interface of KRas4B$^{G12V}$:KRas4B$^{G12V}$, highlighting the residues involved in hydrogen and hydrophobic interactions between two KRas4B$^{G12V}$ molecules. The interface comprises switch I, switch II, and α3 of the two KRas4B$^{G12V}$ molecules. The oncogenic mutation, V12, is annotated in red letters. A schematic representation of the intermolecular contacts predicted by LIGPLOT is presented in the right panel. **(B)** The interacting interface of Rgl2$^{RA}$: Rgl2$^{RA}$, highlighting the residues involved in hydrogen and hydrophobic interactions between two Rgl2$^{RA}$ molecules, shaded in pink and violet. The interaction occurs at the N-terminal between the two anti-parallel β1. A LIGPLOT diagram is shown below.

dimer formation. At 100–250 nM, the peak of the MW reached about 130 kD, which corresponded to the 2:2 hetero-tetramer. The result indicated that, under this experimental condition, the KRas4B$^{G12V}$ and Rgl2$^{RA}$–Halo may form a heterotetramer.

## Discussion

We conducted interaction studies of human KRas4B and Rgl2$^{RA}$ to obtain structural insights into the oncogenic Ras–dependent activation of the RalA/B pathway. Our BLI data show that the oncogenic mutation G12V alters the binding kinetics between KRas4B and Rgl2$^{RA}$. Using X-ray crystallography, we found that KRas4B$^{G12V}$ and Rgl2$^{RA}$ form a heterotetramer and that the oncogenic G12V mutation resides at the interaction surface. Meanwhile, NMR and

mass photometry analyses indicate the complex exists in a heterogeneous status in solution.

### The KRas4B$^{G12V}$:Rgl2$^{RA}$ dimer formed through the KRas4B$^{G12V}$ switch I region displays the common features observed in the canonical mode of Ras-effector interaction

The KRas4B$^{G12V}$:Rgl2$^{RA}$ contacts involving the KRas4B$^{G12V}$ switch I region share the common structural feature as the canonical Ras-effector interface; the β2 in the switch I region of Ras and β2 of RBD/RA interact to form a parallel β-sheet (Fig S8). However, the KRas4B$^{G12V}$ and Rgl2$^{RA}$ molecules at this contact are situated at a slightly different angle compared with Kras:RBD complexes, such as KRas4B$^{WT}$:CRAF$^{RBD}$ (PDB ID 6VJJ) (Fig S8). This is consistent with the recent structural evaluation of all available RAS:RA/RBD complex

**Table 2. NMR structure statistics for Rgl2[a].**

| Quantity | |
|---|---|
| Assigned ¹H/¹³C/¹⁵N chemical shifts | 305/256/56 |
| NOE distance restraints[b] | 866/281/777 |
| Max. distance restraint violation (Å) | 0.13 ± 0.01 |
| Deviations from idealised geometry: | |
| Bond lengths (Å) | 0.0138 ± 0.0001 |
| Bond angles (o) | 1.81 ± 0.03 |
| AMBER energy (kcal/mol) | −3624.25 ± 78.09 |
| AMBER vdW energy (kcal/mol) | −279.71 ± 12.49 |
| Ramachandran plot statistics[c] (%) | 89.5/10.5/0.1/0 |
| Backbone RMSD (Å)[d] | 0.42 ± 0.05 |
| All heavy atom RMSD (Å)[d] | 0.84 ± 0.06 |
| Backbone RMSD to the reference (Å)[e] | 0.91 |
| All heavy atom RMSD to the reference (Å)[e] | 1.68 |

[a]Where applicable, the average value and the SD over the 20 energy-refined conformers obtained by the program OPALp 2.1 in the presence of the experimental restraints. CYANA calculations were started from 100 conformers with random torsion angle values, simulated annealing with 10,000 torsion angle dynamics steps was applied.
[b]Short/medium/long-range distance restraints derived from NOESY spectra.
[c]Percentage of residues in the most favoured/additionally allowed/generously allowed/disallowed regions of the Ramachandran plot according the program PROCHECK.
[d]RMSD to the mean structure for residues 649–657, and 665–732.
[e]RMSD between the closest structure to the reference among the ensemble conformations and the reference structure (the crystal structure of the KRas4B[G12V]:Rgl2 complex).

structures (Eves et al, 2022). In this respect, KRas4B[G12V]:Rgl2[RA] complex belongs to the "RalGDS-family cluster," involving HRas[E31K]:RALGDS[RA] (PDB:1LFD) and KRas[WT/G12V]:Rgl1[RA] (PDB: 7SCW and 7SCX) complexes (Eves et al, 2022), rather than to the "CRAF cluster," suggesting a structure–function correlation.

### The interaction mode between Ras and Rgl2/RALGDS is distinct from other Ras:RBD/RA interactions

The key contributor of the heterotetramer formation seen in our structure of KRas4B[G12V]:Rgl2[RA], and the previously published HRas[E31K]:RALGDS[RA] is the interaction between the switch II region of Ras and the second RBD/RA molecule (Figs 3C and S9A). This feature is distinct from other previously reported Ras:effector crystal structures, where the switch II region is either not participating in the interaction (Ras:CRAF[RBD], PDB ID 6VJJ [Tran et al, 2021]) or is interacting with the same RBD that interacts with the switch I region of the same Ras molecule (Ras:PI3Kγ[RBD] [PDB ID 1HE8] [Pacold et al, 2000], Ras:PLCε[RA] [PDB ID 2C5L] [Bunney et al, 2006], Ras:RASSf5[RA] [PDB ID 3DDC] [Stieglitz et al, 2008]), hence forming a Ras:RA/RBD heterodimer, rather than a heterotetramer (Fig S9B).

One interesting case is a recently reported structural study of Rgl1, yet another RalGEF (Eves et al, 2022). The primary sequences of Rgl1[RA], Rgl2[RA], and RALGDS[RA] are distinct but close to each other when compared with CRAF[RBD], a representative RBD (Fig S10). The

crystal structures of KRas4B[WT] or KRas4B[G12V] in complex with Rgl1[RA] show that Rgl1[RA] interact with the switch I region of Ras in a highly similar manner as other complexes; the similarity is particularly high when compared with KRas4B[G12V]:Rgl2[RA] (this study) and HRas[E31K]:RALGDS[RA] (PDB ID 1LFD) (Figs S9B and S11 top row). In addition, Rgl1[RA] contact the switch II region in a manner similar to PI3Kγ[RBD] and PLCε[RA] (Fig S9B). However, this makes Rgl1[RA] distinct from other RasGEFs, Rgl2[RA], and RalGDS[RA], as these establish the switch II contact using the second RA molecule (Figs S9A and S11).

The crystal structure of Rgl1 in complex with KRas4B[G12V] (PDB ID 7SCX) also suggests a possible heterotetramer formation (Eves et al, 2022). However, the proposed structural arrangements of the KRas4B[G12V]:Rgl1[RA] heterotetramer is distinct from the Ras:Rgl2/RALGDS heterotetramers (Fig S11); in the Ras4B[G12V]:Rgl1[RA] structure, the heterodimer is mediated by an additional Rgl1[RA]:Rgl1[RA] interaction but the two KRas[G12V] molecules are not in contact with each other. Indeed, the second Rgl1[RA] associates with the first Rgl1[RA] through its C-terminal end, which extends into the paired Rgl1[RA] in a reciprocal manner, causing a "domain swap" of the β5 (Eves et al, 2022) (Fig S11). Consequently, the second KRas[G12V] in the KRas[G12V]:Rgl1[RA] heterotetramer is away from the first KRas[G12V] molecule, and the G12V mutation is not directly involved in the molecular contacts. In agreement with this structural arrangement, the affinity between KRas and Rgl1[RA] was reported unaffected by the G12V oncogenic mutation (Eves et al, 2022). This is distinct from the heterotetramer observed in the case of Rgl2[RA], where V12 is directly involved in the RAS:RAS interaction, in agreement with the altered binding kinetics of KRas[G12V] to Rgl2[RA] in comparison to KRas[WT] (Figs 4A, S4, and S11).

### Oligomerisation of KRas:effector complexes

Accumulating evidence suggests that KRas is capable of forming a dimer in solution even in the absence of effectors (Muratcioglu et al, 2015; Jang et al, 2016; Prakash et al, 2017; Sarkar-Banerjee et al, 2017; Lee et al, 2020, 2021; Packer et al, 2021; Andreadelis et al, 2022; Ingolfsson et al, 2022; Ozdemir et al, 2022). The physiological importance of Ras dimerisation and its potential to be a therapeutic target was highlighted by a G12D-specific inhibitor, BI2852, which may cause artificial dimerisation and block the protein function (Kessler et al, 2019; Tran et al, 2020). Furthermore, Ras forms nanoclusters in the membrane in vivo and in vitro (Prior et al, 2003; Plowman et al, 2005; Weise et al, 2011; Zhou et al, 2014; Lakshman et al, 2019). The modes of these Ras oligomer formations are dependent on various parameters, including membrane lipid compositions, Ras nucleotide binding status, availability of Ras effectors, and the actin cytoskeleton and are expected to be context-dependent.

Previously, it has been suggested that Ras in general can form dimers in solution but structural information on these dimers vary and the dimerisation status may therefore be categorised into four classes based on the α helices and β sheets at the interface as follows: (i) α4/α5 (Jang et al, 2016; Prakash et al, 2017; Lee et al, 2020, 2021; Packer et al, 2021; Andreadelis et al, 2022), (ii) α3/α4 (Muratcioglu et al, 2015; Jang et al, 2016; Prakash et al, 2017), (iii) α/β (Lee et al, 2021), and (iv) β/β (Muratcioglu et al, 2015; Jang et al, 2016) (Fig S12). Notably, the KRas4B[G12V]:KRas4B[G12V] interface that we

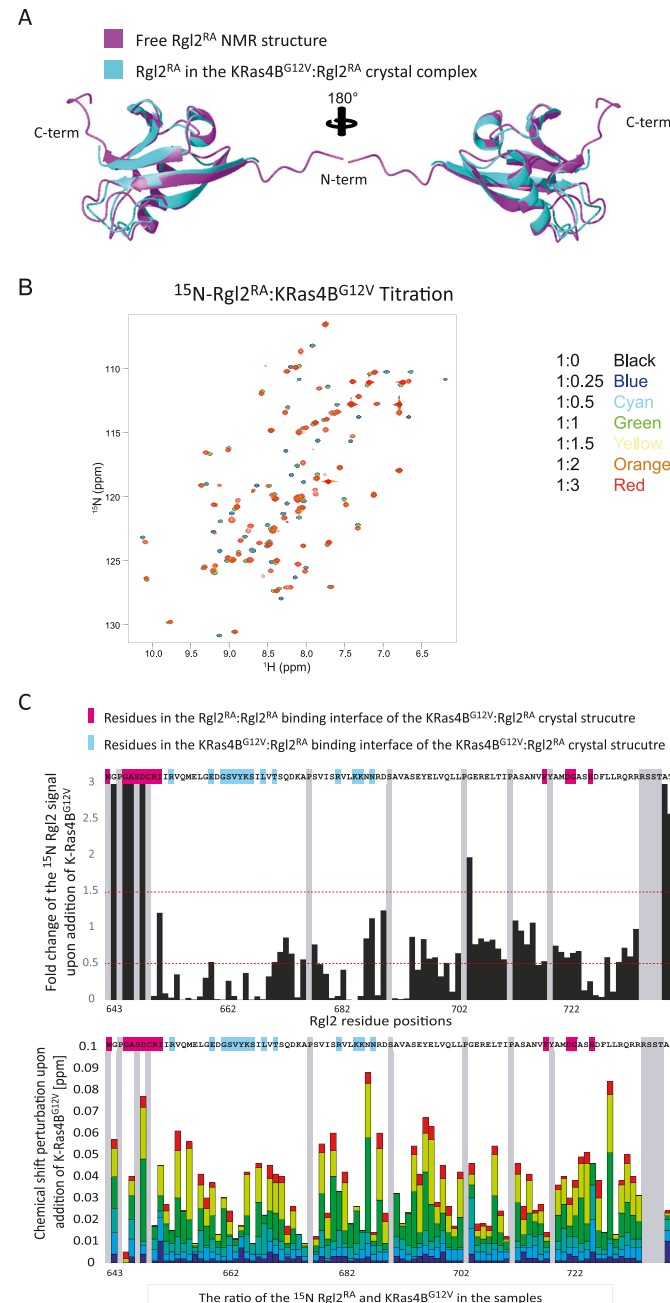

**Figure 5. NMR analysis of KRas4B^G12V:Rgl2^RA complex formation in solution.**
**(A)** Comparison between the Rgl2^RA structures identified in solution NMR (shown in magenta) and in crystal complex with KRas4B^G12V (shown in cyan). **(B)** ^1H-^15N HSQC titration analysis of ^15N-labelled Rgl2^RA upon addition of non-labelled KRas4B^G12V. Overlays of 2D ^1H-^15N HSQC NMR spectra from multipoint titrations of ^15N-labelled Rgl2^RA with non-labelled KRas4B^G12V. Rgl2^RA:KRas4B^G12V molar ratios of the titration samples are colour coded as follows; 1:0 – black, 1:0.25 – blue, 1:0.25 – cyan, 1:0.5 – green, 1:1 – yellow, 1:2 – orange, and 1:3 – red. **(B, C)** The NMR signal intensity changes (upper panel) and the chemical shift perturbation (lower panel) of backbone ^1H^N and ^15N nuclei of Rgl2^RA with non-labelled KRas4B^G12V, presented in (B), are summarised as column diagrams as a function of Rgl2^RA amino acid sequence. Proline and unassigned Rgl2^RA residues are shaded in grey. Rgl2^RA residues involved in the Rgl2^RA:Rgl2^RA interface of the KRas4B^G12V:Rgl2^RA crystal structure are highlighted in pink on the amino acid sequence, and residues involved in the KRas4B^G12V:Rgl2^RA interface are highlighted in blue. Rgl2^RA residue position numbers, according to the UniProt, are indicated at

observe in the KRas4B^G12V:Rgl2^RA heterotetramer does not belong to any of these categories and uses the unstructured regions of switch I, switch II, V12 in the P-loop and K88 (Figs 4A and S12), in a similar manner to the HRas^E31K:RALGDS^RA crystal (PDB ID 1LFD) (Huang et al, 1998). Although this mode of tetramer formation could be an outcome of a crystal-packing artefact (Vetter et al, 1999), it is worth noting that the spatial arrangement of symmetry mates of human HRas^WT crystal (PDB ID 5P21) also shows a similar Ras:Ras contact (Pai et al, 1990) (Fig S13C). Therefore, this Ras:Ras contact mode may reflect the intrinsic nature of Ras molecules, which may be stabilized by both KRas4B 12V and Rgl2^RA to form the heterotetramer.

Our mass photometry analysis supports a tetramer formation in solution. However, the rotational correlation time from the NMR relaxation experiments was rather close to that estimated from the size of the heterodimer. Furthermore, our SEC-MALS attempt only detected a heterodimer formation (data not shown), indicating that the heterotetramer formation may involve certain environmental conditions or an exchange process, as was the case for the KRas^GV:Rgl1^RA where tetramerization was observed by SEC only on 6-mo old samples (Eves et al, 2022). Physiological settings, including the presence of the plasma membrane and the full-length effector molecules rather than just the RBD/RA, may stabilize higher-order complex formations.

## Possible involvement of V12 in the KRas4B:Rgl2^RA complex formation and oncogenicity

The KRas4B^G12V:KRas4B^G12V interface of the KRas4B^G12V:Rgl2^RA tetramer complex involves the residue V12, an oncogenic amino-acid substitution occurring in about 28% of mutated *KRAS* cases (COSMIC). This G12V substitution alters the binding kinetics towards Rgl2^RA compared with the wildtype (Fig 1B). In the KRas4B^G12V:Rgl2^RA tetramer complex, the presence of V12 creates a larger hydrophobic pocket together with Y32, compared with the wildtype case of G12, seen in the HRas^WT crystal (PDB ID 5P21) or in the HRas^E31K:RALGDS^RA crystal (PDB ID 1LFD) (Fig S13A–C). Interestingly, the E31K substitution, used to stabilize the complex in the HRas^E31K:RALGDS crystal structure (Huang et al, 1998) (Fig S13B), has been reported in cancer samples with *HRAS* mutations (COSMIC). Therefore, the capability of Ras mutants to form a stable complex with RalGEFs may be directly linked with Ras oncogenicity. It is interesting to note that in the previously proposed Ras:CRAF heterotetramer complex derived from SAXS data (Packer et al, 2021), the position of residue 12 is relatively distant from the interacting surfaces. An interesting speculation can be that the oncogenic substitution mutations at glycine 12 might have a greater

the bottom of the diagram. (Upper panel) The signal intensities of Rgl2^RA residues in the presence of three times molar excess of KRas4B^G12V were divided by the signal intensities in the absence of KRas4B^G12V and plotted as a bar chart graph. Red-dotted lines are drawn at the fold-change values of 0.5 and 1.5 to highlight the residues that show a substantial increase or decrease of the signals upon the addition of KRas4B^G12V. (Lower panel) The chemical shift perturbation of backbone ^1H^N and ^15N nuclei of Rgl2^RA with non-labelled KRas4B^G12V. The mean shift difference $\Delta\delta_{ave}$ was calculated as $([\Delta\delta^1H^N]^2 + [\Delta\delta^{15}N/10]^2)^{1/2}$ where $\Delta\delta^1H^N$ and $\Delta\delta^{15}N$ are the chemical shift differences between Rgl2^RA on its own and in the presence of non-labelled KRas4B^G12V. The bar graphs are colour coded according to the Rgl2^RA–KRas4B^G12V concentration ratio.

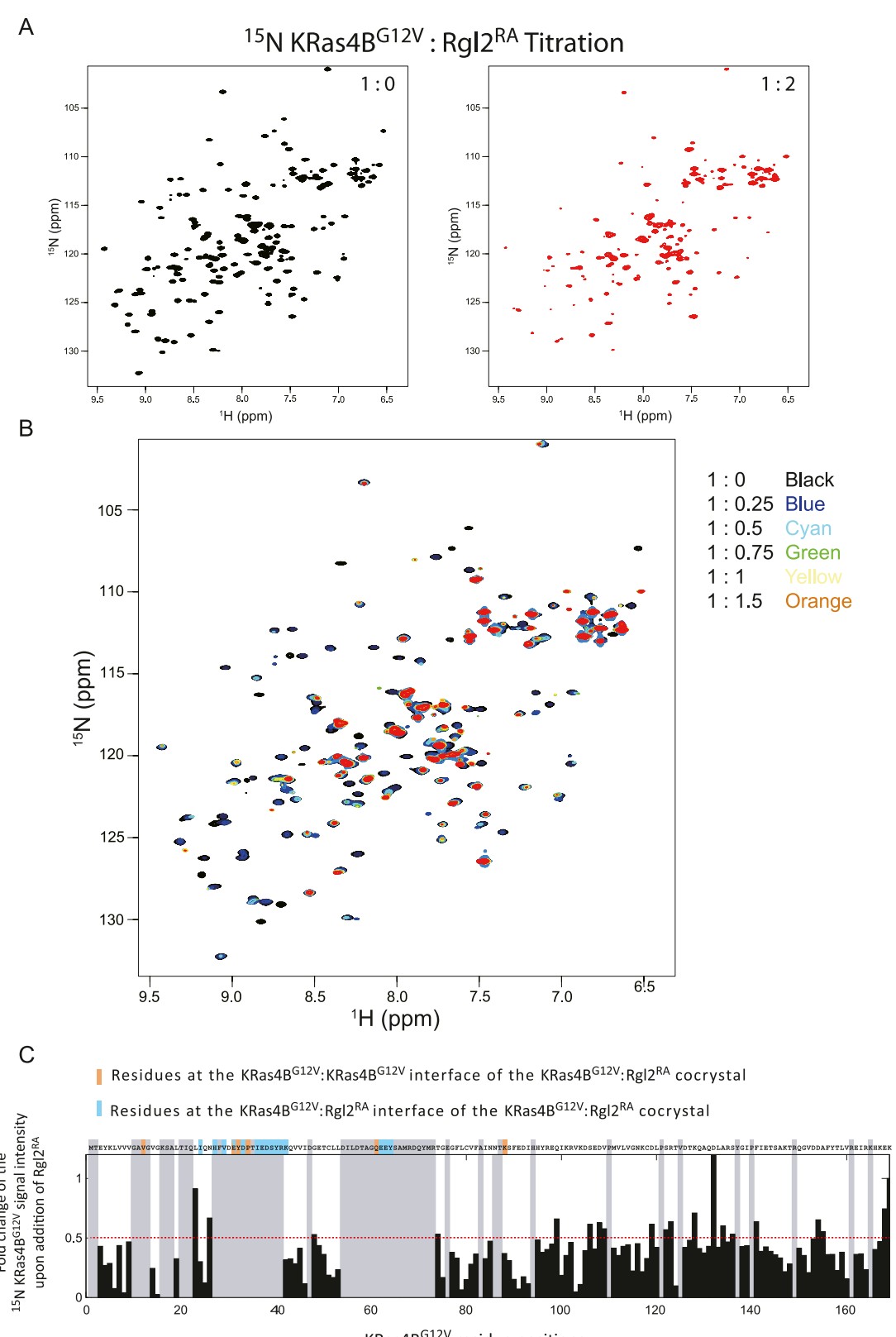

**Figure 6. KRas4B^{G12V}:Rgl2^{RA} complex formation in solution.**
[1]H-[15]N-HSQC titration analysis of [15]N-labelled KRas4B^{G12V} upon addition of non-labelled Rgl2^{RA} supports the KRas4B^{G12V}:Rgl2^{RA} tetramer formation. **(A, B)** The 2D [1]H-[15]N-HSQC NMR spectra of KRas4B^{G12V}. [15]N-labelled KRas4B^{G12V} was titrated with non-labelled Rgl2^{RA}. **(A)** The 2D [1]H-[15]N-HSQC spectra of KRas4B^{G12V}: Rgl2^{RA} complex mixed with

impact on RalGEF-mediated signalling than Raf kinase-mediated signalling, as suggested by previous studies through real-time parallel NMR analyses (Smith & Ikura, 2014). The differences in the KRas4B^G12V:Rgl2^RA crystal structure (this study) and the Ras:CRAF heterotetramer (Packer et al, 2021) provide possible structural explanations for this observation.

To summarize, our work demonstrates an altered binding kinetics of KRas4B^G12V oncogenic mutant with a RalGEF, Rgl2. The G12V mutation resides at the interface of the KRas4B^G12V-Rgl2^RA co-crystal complex. The information may open the way to target oncogenic *KRAS*–induced tumorigenesis by novel strategies, including interfering molecules for the newly identified interfaces.

# Materials and Methods

### Plasmid constructs

The RA of human Rgl2 was obtained by amplifying a cDNA fragment encoding the position 643–740 of the human Rgl2 by PCR with a pair of primers (5′ TACTTCCAATCCATGGGGCCAGGGGCCTCTGATTGCCG 3′) and (5′ TATCCACCTTTACTGTCA TGTAGCAGTAGAGGACCTTCGCCGCTGC 3′) using human cDNA prepared from hTERT RPE-1 cells using GoScript Reverse Transcription System (Promega) following the manufacturer's instruction. The amplified Rgl2^RA fragment was cloned into pLEICS2 vector (PROTEX, University of Leicester), which contains a GST affinity tag and a tobacco etch virus (TEV) protease cleavage site at the N-terminal end of the Rgl2^RA, using In-Fusion HD EcoDry enzyme (#638915; Takara Bio), following the manufacturer's instruction.

A cDNA fragment encoding the position 1–169 (C-terminal truncated) of the human wildtype KRas4B isoform was amplified by PCR with a pair of primers (5′ TACTTCCAATCCATG ACTGAATA-TAAACTTGTGGTAGTTGGAGCTG 3′) and (5′ TATCCACCTTTACTGTCA CTTTTCTTTATGTTTCGAATTCTCGAACTAATGTATAG 3′) using human cDNA prepared from hTERT RPE-1 cells as described above. The produced fragment was cloned in pLeics1 plasmid (PROTEX, University of Leicester), which introduces a His6 tag and a TEV cleavage site at the N-terminus. Site-directed mutagenesis was conducted to introduce the oncogenic G12V mutation using a pair of DNA oligos (5′ AGTTGGAGCTGTTGGCGTAGGCAAGAGTGCC 3′) and (5′ GTCAAGG-CACTCTTGCCTACGCCAACAGCTCCAACTAC 3′) by following the Quik-Change method (Agilent Technologies).

The RBD of human BRAF was obtained by amplifying a cDNA fragment encoding the position 151–232 of the human BRAF by PCR with a pair of primers (5′ TACTTCCAATCCATGTCACCACAAAAACCTATCGTTAGAGTCTTCCTGCC

3′) and (5′ TATCCACCTTTACTGTCAAAGTGGAACATTCTCCAACACTTCCACATGCAATTC 3′) using human cDNA prepared from hTERT RPE-1 cells as described above. The produced fragment was cloned into pLEICS2 vector (PROTEX, University of Leicester) as described above.

The Rgl2^RA–Halo construct was prepared using the above-described GST–Rgl2^RA construct by amplifying the DNA fragment encoding GST–Rgl2^RA by PCR using a pair of primers primers (5′ AGGAGATATACATATGTCCCCTATACTAGGTTATTGGAAAATTAAGGG 3′) and (5′ CAGTACCGATTTCGGATCCTGTAGCAGTAGAGGACCTTCGCCG 3′) and inserting the PCR product into pLEICS90 vector (PROTEX, University of Leicester) that adds a Halo tag (Los et al, 2008) at the C-terminal end.

### Protein expression

DE3 Rosetta cells (Novagen) carrying expression plasmids were grown at 37°C in TY media until OD600 reached about 0.6. Then protein expression was induced by adding isopropyl β-D-1-thiogalactopyranoside to a final concentration of 0.5 mM and keeping the culture at 18°C overnight in a shaking incubator. Cells were collected by centrifugation and resuspended in either Sl1 buffer (Tris 20 mM [pH 7.65], NaCl 150 mM, 5 mM imidazole) for His-tagged KRas or in PBS–NaCl buffer (237 mM NaCl, 2.7 mM KCl, 8 mM $Na_2HPO_4$, and 2 mM $KH_2PO_4$, pH 7.4) for Rgl2^RA and BRAF^RBD. The cell suspensions were then stored at –80°C.

### Protein expression of stable isotope labelling KRas4B^G12V and RA of Rgl2 for NMR measurement

The gene encoding human KRas4B^G12V and Rgl2^RA were constructed into the expression vector pGHL9 and pLEICS2 over-expressed in *Escherichia coli* (*E.coli*) strain BL21 (DE3) and Rosetta, respectively. Uniformly, ^13C, ^15N-labeled protein was obtained by growing bacteria at 37°C in M9 minimal media, containing (^13C_6)-glucose and ^15NH_4Cl (Isotec) as the sole carbon and nitrogen source, supplemented with 20 mM $MgSO_4$, 0.1 mM $CaCl_2$, 0.4 mg/ml thiamin, 20 μM $FeCl_3$, salt mix (4 μM $ZnSO_4$, 0.7 μM $CuSO_4$, 1 μM $MnSO_4$, 4.7 μM $H_3BO_3$), and 50 mg/l ampicilin. KRas4B^G12V NMR sample was prepared essentially as described previously (Ito et al, 1997). Protein expression of Rgl2^RA was induced by adding 119 mg/l isopropyl β-D-1-thiogalactopyranoside at an OD 600 nm of 0.5. After 18 h of further growth, cells were harvested and washed with a pH 7.5 lysis buffer (50 mM Tris–HCl, 25% sucrose, and 0.01% NP-40). Uniformly ^15N-labeled KRas4B^G12V and Rgl2^RA were produced by the identical

the molar ratio of 1:0 (black, left panel) and 1:2 (red, right panel) are shown. Many signals from ^15KRas4B^G12V residues disappeared upon the addition of Rgl2^RA. **(B)** Superimposed 2D ^1H-^15N-HSQC NMR spectra of ^15N-labelled KRas4B^G12V:Rgl2^RA titration experiments. The titration samples are colour coded as follows; 1:0 – black, 1:0.25 – blue, 1:0. 5 – cyan, 1:0.75 –green, 1:1 – yellow, 1:1.5 – orange, and 1:2 – red. **(C)** Fold changes of the signal intensities of ^15N-labelled KRas4B^G12V upon the addition of non-labelled Rgl2^RA. The signal intensities of KRas4B^G12V residues in the presence of two times molar excess of Rgl2^RA were divided by the signal intensities in the absence of Rgl2^RA, and the obtained values were plotted as a column graph. Undetectable residues are shaded in grey. The residue T2 was also shaded grey, as the chemical shift after the addition of Rgl2^RA overlapped with other signals. A red-dotted line is drawn at the fold-change values of 0.5 to indicate that most residues show a substantial decrease in the signals upon the addition of Rgl2^RA. KRas4B^G12V residues in the KRas4B^G12V:KRas4B^G12V interface of the KRas4B^G12V:Rgl2^RA crystal structure is highlighted in orange, and residues in the KRas4B^G12V:Rgl2^RA interface of the crystal structure is highlighted in blue. KRas4B^G12V residue positions according to the UniProt are indicated at the bottom of the diagram.

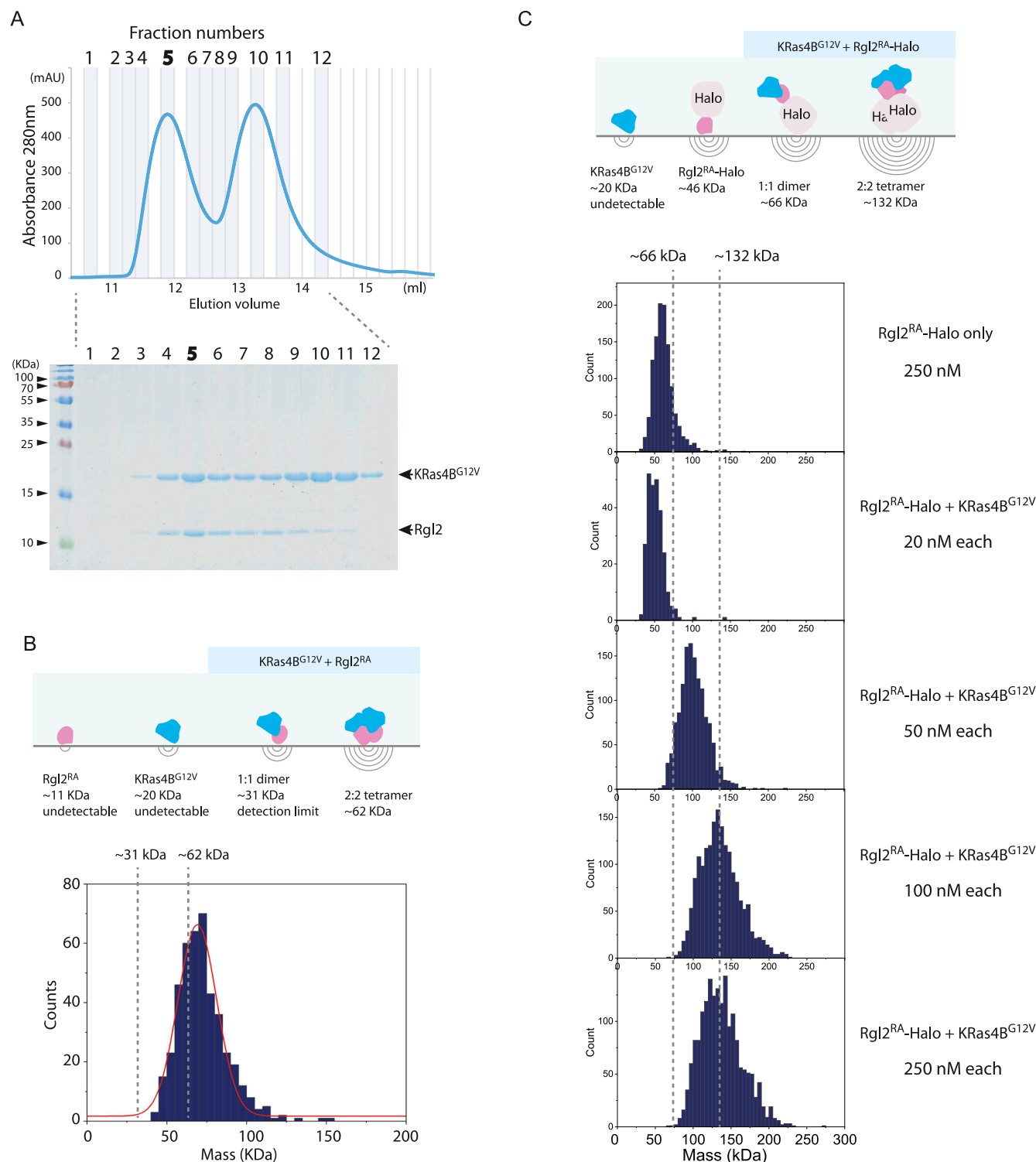

**Figure 7. KRas4B$^{G12V}$:Rgl2$^{RA}$ complex can form a heterotetramer in solution.**
Mass photometry analysis of KRas4B$^{G12V}$:Rgl2$^{RA}$ complex in solution. **(A)** KRas4B$^{G12V}$:Rgl2$^{RA}$ complex was purified using size exclusion chromatography. The fractions were analysed with 15% SDS–PAGE gel (upper panel) according to the elution profile (lower panel). Fraction 5 was chosen and measured using mass photometry (OneMP, Refeyn). **(B)** The cartoon representing the different complex configurations possibilities indicates that only heterodimers (~31 kD) and heterotetramers (~62 kD) can be detected by the system. Histogram of the frequency counts against the purified KRas4B$^{GV}$:Rgl2$^{RA}$ complex with fitting for Gaussian distribution (red). The duration of the video analysed was 60 s, and it shows that the population identified has an average mass of ~68 kD, which is in good agreement with the expected MW of the heterotetramer (61.7 kD). **(C)** KRas4B$^{G12V}$ and Halo-tagged Rgl2$^{RA}$ (Rgl2$^{RA}$–Halo) complex formation. The cartoon represents MWs of possible complexes. KRas4B$^{G12V}$ and Rgl2$^{RA}$–Halo were separately prepared and were mixed to generate a premix 2 µM sample, which was further diluted to 20, 50, 100, and 250 nM before the measurement.

steps unless growing cells in M9 medium containing ($^{12}C_6$)-glucose and $^{15}NH_4Cl$ (Isotec).

## Purification of GST-tagged Rgl2[RA], Rgl2[RA]-Halo, and BRAF[RBD]

Bacteria cell suspensions were thawed and supplemented with Triton X-100 to a final concentration of 0.1% (vol/vol). Cells were broken by a probe sonicator, and insoluble materials were removed by centrifugation. The supernatant was mixed with glutathione (GSH) beads (#17-0756-01; GE Healthcare) and incubated for 20 min at 4°C. The GSH beads were washed three times with the PBS–NaCl buffer. The GST–Rgl2[RA] and GST–BRAF[RBD] fusion proteins were eluted by the elution buffer (50 mM Tris-Cl [pH 8.0], 100 mM NaCl, 5 mM GSH) for BLI experiments. The Rgl2[RA] and Rgl2[RA]–Halo were separated from the GST tag through cleavage by TEV protease, which was prepared as previously described (Kapust et al, 2001; Tropea et al, 2009), for structural analysis and mass photometry analysis. The obtained Rgl2[RA], Rgl2[RA]–Halo, or GST–Rgl2[RA]/GST–BRAF[RBD] fusion proteins were concentrated using a concentrator (10 KD MWCO, Amicon Ultra centrifugal filters, #UFC901024; Merck) and filtrated using a centrifugal filter unit (Ultrafree, #UFC30GV00; Millipore) before conducting SEC with a gel-filtration column (HiLoad Superdex 75; GE Healthcare) attached to an FPLC system. SEC was carried out in the gel filtration (GF) buffer (20 mM Tris-Cl [pH 7.65], 100 mM NaCl, 5 mM $MgCl_2$, 1 mM Tris (2-carboxyethyl) phosphine [TCEP]).

## Purification of His-tagged KRas4B wildtype and oncogenic G12V mutant proteins

Bacteria cell suspensions were supplemented with Triton X-100 to a final concentration of 0.1% (vol/vol). Cells were broken by a probe sonicator and insoluble materials were removed by centrifugation. The soluble cell lysates were applied on an Ni-sepharose excel (#17-3712-01; GE Healthcare), packed in a column of 4 ml bed volume with Sl1 buffer buffer (20 mM Tris, pH 6.5, 150 mM NaCl, 5 mM imidazole). The column was washed with 20 ml of SL1 buffer, then with 20 ml of Sl3 buffer (20 mM Tris, pH 6.5, 150 mM NaCl, 6 mM imidazole) and finally with 15 ml of Sl4 buffer (20 mM Tris, pH 6.5, 150 mM NaCl, 10 mM imidazole). The His-tagged KRas protein was eluted from the column by applying 10 ml of elution buffer (50 mM Tris, pH 7.65, 150 mM NaCl, 200 mM Imidazole), followed by 10 ml of 1 M imidazole. To remove the His6-tag at the N-terminal end, TEV protease, prepared as previously described (Tropea et al, 2009), was added to the elution fraction to about a 2% molar ratio of the His6–KRas4B preparation and incubated overnight at 4°C. The cleaved KRas samples were further purified by SEC using a gel-filtration column (HiLoad Superdex 75; GE Healthcare) in the GF buffer (20 mM Tris–Cl [pH 7.65], 100 mM NaCl, 5 mM MgCl2, 1 mM TCEP). The purified KRas (WT) and KRas(GV) concentrations were determined by absorbance at 280 nm. The extinction coefficient ($\varepsilon$) for KRas (WT) and KRas(GV) was estimated to be 19,685 cm-$^1$ $M$ $^{-1}$, by taking into account that the

bound GTP adds 7765 cm-$^1$ $M$ $^{-1}$ (Smith & Rittinger, 2002), and the molecular weights (including GTP) were estimated to be 19,856 and 19,898, respectively. Nucleotide exchange of the purified KRas4B wildtype or G12V proteins was carried out essentially as previously described (Ito et al, 1997). The proteins were diluted 10 times by the exchange buffer (20 mM Tris–Cl [pH 7.5] 1 mM EDTA, 1 mM TECP), and the sample was supplemented with EDTA to a final concentration of 5 mM. The sample was mixed with about 10 time molar excess of guanosine 5'-[$\beta,\gamma$-imido]triphosphate (GMMPPNP, G0635; Sigma-Aldrich) or 5'-GTP (G9977; Sigma-Aldrich) or 5' –GDP (G7127; Sigma-Aldrich). The reaction was incubated at 37°C for 20 min then put it on ice for 20 min. The protein was terminated by adding ice cold $MgCl_2$ to a final concentration of 20 mM. The excess nucleotides were removed by SEC as described below.

## Purification of isotope-labelling KRas4B[G12V]

All the procedures described below were carried out at 4°C unless otherwise stated. All the isotope-labelled KRas4B[G12V] samples were purified by the same step. The cells dispersed in the lysis buffer were disrupted by sonication for 30 min on ice with hen egg lysozyme (0.1 mg/ml). The cell debris was clarified by centrifugation at 14,000$g$ for 1 h. The supernatant was loaded onto a 25 ml of DEAE-Sepharose Fast Flow (Cytiva) anion exchange equilibrated with buffer A (50 mM Tris–HCl [pH 7.5], 1 mM $MgCl_2$, 1 mM DTT, 0.1 mM APMSF [FUJIFILM Wako]). After washing the column with buffer A until sufficiently low of UV absorption at 280 nm, the KRas4B[G12V] protein was eluted by linearly increasing the concentration of KCl from 0 to 350 mM with a flow rate of 0.5 ml/min in buffer A. The fractions containing the target protein were concentrated to 5 ml with Amicon Ultra-15 10 kD (Merck). The concentrated sample was loaded onto a 320 ml of HiLoad Superdex 75 (GE Healthcare Life Science) gel filtration with a flow rate of 0.8 ml/min using FPLC systems (AKTA pure 25; GE Healthcare Life Science). The 5 ml sample concentrated from the fractions involving the target proteins with Amicon Ultra-15 10 kD was loaded on Resource Q (GE Healthcare Life Science) anion-exchange column equilibrated with buffer A using the FPLC systems. After washing the column with 30 ml of buffer A, KRas4B[G12V] was eluted by KCl, the KRas4B[G12V] protein was eluted by linearly increasing the concentration of KCl from 0 to 350 mM with a flow rate of 1 ml/min in buffer A. The purification of isotope labelled Rgl2[RA] was performed by the same step described above. The purity of the KRas4B[G12V] and Rgl2[RA] samples in each step was confirmed by SDS–PAGE. Protein concentrations were determined by Nano Drop 2000 (Thermo Fisher Scientific) measuring UV absorption at 280 nm. KRas4B[G12V] samples for NMR measurements were concentrated and dissolved in NMR buffer A (90% $^1H_2O$/10% $^2H_2O$ containing 20 mM Tris–HCl [pH 7.5], 100 mM NaCl, 5 mM $MgCl_2$, 1 mM $\beta$-mercaptoethanol). Rgl2[RA] samples for NMR measruments were NMR buffer B (90% $^1H_2O$/10% $^2H_2O$ containing 1 mM $Na_2HPO_4$-$NaH_2PO_4$ [pH 7.4], 150 mM NaCl).

---

At 250 nM, Rgl2[RA]–Halo without KRas4B[G12V] showed about 55 kD, within the range of the predicted MW (~46 kD). In contrast, the mixed samples showed an increase in the MW as the concentration was increased. At 50 nM, the observed MW was about 90 kD, which may represent the predicted dimer, and at 100–250 nM, the observed MW peaked at about 130 kD, which coincided with the predicted MW for the tetramer.

## KRas4B–Rgl2[RA] and KRas4B–BRAF[RBD] binding measurements using BLI

Octet R8 (Sartorius) was used for BLI assays of KRas4B (G12V or WT) and Rgl2[RA] and BRAF[RBD] interactions. Anti-GST biosensors (#18-5096; Sartorius) were used to immobilise GST-Rgl2[RA] (provided as a 0.7 $\mu$M solution in the binding reservoir well, the concentration was determined by absorbance at 280 nm using the following values; the extinction coefficient ($\varepsilon$) 48,820 cm$^{-1}$ M$^{-1}$, and the molecular weight 37,495.18), GST–BRAF[RBD] (encoding BRAF a.a. 151–232, N-terminally fused with GST, provided as a 0.7 $\mu$M solution in the binding reservoir well, the concentration was determined by absorbance at 280 nm using the following values; the extinction coefficient ($\varepsilon$) 56,840 cm$^{-1}$ M$^{-1}$, and the molecular weight 36,024.95). The baseline was stabilised in GF buffer (20 mM Tris, pH 7.65, 100 mM NaCl, 5 mM MgCl$_2$, 1 mM TCEP) for 200 s. As a negative control, GST only (provided as a 0.7 $\mu$M solution in the binding reservoir well) was used. The association of KRas4B was measured for 400 s in five or six serial dilutions concentrations. For Rgl2[RA] binding, the KRas concentrations ranged from 200 nM to 15 $\mu$M, and for BRAF[RBD] binding, the KRas concentrations ranged from 20 nM to 5 $\mu$M. The dissociation steps were measured in fresh GF buffer for 400 s. For each assay, a biosensor immobilised with GST only and a sample well with only buffer instead of KRas4B (WT/G12V) was set up for double referencing. The experiments were conducted at 20°C. The resulting data were processed using Octet Analysis Studio (ver. 13.0) (Sartorius), and reference biosensor (loaded with GST-only) and reference wells (containing no KRas) were subtracted from sample wells (double reference).

## Analysis of bound nucleotide

The nucleotide-binding status of KRas4B[WT] and KRas4B[G12V] were examined by denaturing the proteins and detecting the released nucleotides, essentially following the previous studies (Smith & Rittinger, 2002). About 2 nmoles of KRas4B molecules were adjusted to the volume of 200 $\mu$l with GF buffer. Add 12.5 $\mu$l of 10% perchloric acid to denature and precipitate the protein. The supernatant was neutralised either by 8.75 $\mu$l of 4 M CH$_3$COONa, pH 4.0, or 12 $\mu$l of 1 M Tris–Cl (pH 8.8). The sample was centrifuged again, and the supernatant was analysed by HPLC using an ion exchange column (Partisil 10 SAX column; Whatman). For the result presented in Fig 1A, the chromatography was run using 0.6 M NH$_4$H2PO$_4$ (buffer B) at a flow rate of 0.8 ml/min. For the result presented in Fig 1D, the column was first equilibrated with 10 mM NH$_4$H2PO$_4$ (buffer A), and the column was run under the following gradient condition with 0.6 M NH$_4$H2PO$_4$ (buffer B) at a flow rate of 0.8 ml/min. Step 1: 100% buffer A (0% buffer B) for 11 min, step 2: a gradient increase to 40% buffer B over 6 min, step 3: a gradient increase to 50% buffer B over 23 min, step 4: a gradient increase to 100% buffer B over 1 min, step 5: 100% buffer B for 19 min, step 6: a gradient decrease to 0% buffer B (100% buffer A) over 1 min, and Step 7: 100% Buffer A for 14 min. The nucleotides were detected by 254 nm absorption. As a reference control, 1 $\mu$l of 10 mM GTP or GDP was diluted to 200 $\mu$l GF buffer and was processed in the same manner as protein samples.

## Crystallography

The purified and GMPPNP-loaded KRas4B (G12V) and Rgl2[RA] were mixed in the GF buffer and the complex was purified on SEC using a gel-filtration column (HiLoad Superdex 75; GE Healthcare). The peak fractions containing both proteins in the 1:1 ratio were collected and concentrated to set up crystallization screenings. Crystals of KRas4B (G12V) and Rgl2[RA] were obtained using sitting-drop vapour diffusion at room temperature, with 100 nl of protein (11.6 mg/ml) against 100 nl of crystallisation buffer (0.2 M sodium/potassium phosphate, ph 7.5, 0.1 M HEPES, pH 7.5, 22.5% vol/vol PEG Smear Medium [Molecular Dimensions MD2-100-259], 10% vol/vol glycerol). The crystals were frozen in liquid nitrogen with 20% glycerol as cryoprotectant. Data were collected at Diamond beamline I04. AIMLESS (Evans & Murshudov, 2013) was used for data reduction before obtaining Phaser solution using the HRas–RALGDS[RA] complex structure (PDB ID 1LFD) as search model with PhaserMR (McCoy et al, 2007). The structure was built using multiple rounds of refinements using PDBredo, REFMAC, PHENIX, and COOT (Emsley et al, 2010; Murshudov et al, 2011; Joosten et al, 2014; Torices & Munoz-Pajares, 2015). The coordinates of the complex have been deposited to the Protein Data Bank (PDB) under access code 8B69.

## Circular dichroism (CD) spectroscopy

KRas4B(WT) and (G12V) proteins at a concentration of 20 $\mu$M were prepared in the CD buffer (50 mM phosphate [pH 7.6], 1.5 mM Tris [pH 7.6], 5 mM MgSO4, 7.5 mM NaCl, 0.375 mM MgCl2), placed in a quartz cuvette of 0.1 cm path length, and CD spectra were recorded at wavelengths ranging from 195 to 250 nm using a Chirascan-plus CD spectrometer (Applied Photophysics) at 20°C. The melting curves of these proteins were examined at 220 nm at temperatures ranging from 20°C to 90°C. Measurements were conducted at 1°C increment.

## NMR spectroscopy

KRas4B[G12V] NMR sample was prepared essentially as described previously (Ito et al, 1997). The bacteria expression plasmid for HRas was modified to encode KRas4B[G12V] by site-directed mutagenesis and used to produce KRas4B[G12V]. Loading of a GTP analogue, GMPPNP (Jena Bioscience) was conducted essentially as previously described (Ito et al, 1997). Rgl2[RA] sample was prepared as described above. All NMR samples were measured in 20 mM Tris–Cl (pH 7.65), 100 mM NaCl, 5 mM MgCl$_2$, and 0.1% $\beta$-mercaptoethanol at 303 K. All spectra were analysed with the CcpNmr Analysis 2.5.1 software. Backbone chemical shifts of KRas4B[G12V] and Rgl2 have been deposited to the BioMagResBank with accession ID 34754 and PDB with accession ID 8AU4, respectively.

All NMR experiments were performed at 25°C probe temperature in a triple-resonance cryoprobe fitted with a z-axis pulsed field gradient coil, using Bruker AVANCE III HD 600 MHz spectrometers. All spectra were processed with the Azara software package (Boucher, 2010). For the 3D data, the two-dimensional maximum entropy method (2D MEM) or quantitative maximum entropy (Hamatsu et al, 2013) was applied to obtain resolution enhancement for the indirect dimensions. All NMR spectra were visualized and analyzed

using the CcpNmr Analysis 2.5.0 software (Vranken et al, 2005). All of the 3D triple-resonance experiments used for the assignments of KRas4B and Rgl2[RA] were performed on $^{13}C/^{15}N$ samples in NMR buffer A and B, respectively. The backbone $^1H^N$, $^{13}C'$, and $^{15}N$ for KRas4B and Rgl2[RA], and side-chain $^{13}C^\alpha$ and $^{13}C^\beta$ resonance assignments for Rgl2[RA] were achieved by analyzing six types of 3D triple-resonance experiments, HNCO, HN(CA)CO, HNCA, HN(CO)CA, CBCANNH, and CBCA(CO)NNH. 3D HBHA(CBCACO)NH, H(CCCO)NH, (H) CC(CO)NH, HCCH-COSY, and HCCH-TOCSY experiments on the $^{13}C/^{15}N$ -labeled Rgl2[RA] were performed for side-chain $^1H$ and $^{13}C$ resonance assignments. A 15 ms $^{13}C$ isotropic mixing time was used for the (H)CC(CO)NH, H(CCCO)NH, and HCCH-TOCSY experiments. For the collection of NOE-derived distance restraints of Rgl2[RA], 3D $^{15}N$-separated and 3D $^{13}C$-separated NOESY-HSQC spectra were measured on the $^{13}C/^{15}N$ -labeled Rgl2[RA]. A 100 ms NOE mixing period was used for the 3D NOESY experiments. All 2D and 3D NMR data were recorded using the States-TPPI protocol for quadrature detection in indirectly observed dimensions. Water flip-back $^1H$ pulses and the WATERGATE pulse sequence were used for solvent suppression in the experiments performed on $^{15}N$-labeled and $^{13}C/^{15}N$-labeled samples, whereas presaturation and gradient-spoil pulses were used for $^{13}C$-labeled samples.

A series of 2D $^1H$-$^{15}N$ HSQC spectra were measured for titration experiments of $^{15}N$-labelled KRas4B[G12V] within the presence of non-labelled Rgl2[RA]. The experiments were performed in the NMR buffer at 25°C, and the peptide concentration was increased stepwise (for the $^{15}N$-KRas4B[G12V]/Rgl2[RA], its molar ratio of 1:0.25, 1:0.5, 1:0.75, 1:1, 1:1.5, and 1:2 were used, whereasfor the $^{15}N$-Rgl2[RA]/KRas4B[G12V], its molar ratio of 1:0.25, 1:0.5, 1:1, 1:1.5, 1:2, and 1:3 were used). The mean chemical shift difference $\Delta\delta_{ave}$ for each amino acid was calculated as $([\Delta\delta^1H^N]^2 + [\Delta\delta^{15}N]^2)^{1/2}$ where $\Delta\delta^1H^N$ and $\Delta\delta^{15}N$ are the chemical shift differences (Hz) between KRas4B[G12V] or Rgl2[RA] on their own and the proteins in the presence of the other side.

## NMR structure calculation

Intra-residual and long-range NOEs were automatically assigned by the program CYANA with the use of automated NOE assignment and torsion angle dynamics (Güntert & Buchner, 2015). The peak position tolerance was set to 0.03 ppm for the 1H dimension and to 0.3 ppm for the 13°C and 15 N dimensions. Hydrogen-bond and dihedral angle restraints were not used. CYANA calculations were started from 100 conformers with random torsion angle values, simulated annealing with 50,000 torsion angle dynamics steps was applied. The 20 conformers with the lowest final target-function values of CYANA were selected and optimised with OPALp 2.1 (Luginbühl et al, 1996; Koradi et al, 2000) using the AMBER force field (Cornell et al, 1995; Ponder & Case, 2003).

## NMR relaxation experiment

The longitudinal relaxation times ($T_1$), the transverse relaxation times ($T_2$), and the steady-state heteronuclear {$^1H$}-$^{15}N$ NOEs were measured at 25°C using uniformly $^2H/^{13}C/^{15}N$-labeled KRas4B[G12V] with non-labelled Rgl2[RA] on a Bruker Avance III HD 600 MHz spectrometer equipped with a cryogenic H/C/N triple-resonance probe head. Each experiment was acquired in a pseudo 3D manner.

Eight relaxation delays in the range 20–1,800 ms and eight delays between 8.7–139 ms were used for the $T_1$ and $T_2$ experiments, respectively. NOE ratios were obtained from intensities in experiments recorded with (1 s relaxation delay followed by 10 s saturation) and without (relaxation delay of 11 s) saturation. The overall rotational correlation time, effective correlation times, and order parameters were obtained by the program relax 5.0. with spectral density functions as defined by the Lipari-Szabo model-free approach. A theoretical rotational correlation time ($\tau_c$) was calculated by the Stokes-Einstein-Debye relation,

$$\tau_c = \frac{4\pi\eta r_{eff}^3}{3k_B T}$$

where $\eta$ is the viscosity, $r_{eff}$ is the effective hydrodynamic radius of a molecule, $k_B$ is the Boltzmann constant, and $T$ is the temperature. The rotational correlation time of the KRas4B[G12V]: Rgl2[RA] complex was estimated with $T = 298.0$ K, $\eta = 0.890$ mPa, and $r_{eff} = 27.1$ Å.

## Mass photometry measurement

For KRas4B[G12V] and Rgl2[RA] complex formation, freshly purified KRas4B[G12V] and Rgl2[RA] were mixed in GF buffer, concentrated to about 500 μl using a concentrator (Amicon Ultra centrifugal filters, #UFC901024; Merck), and loaded to a gel-filtration column (HiLoad Superdex 75; GE Healthcare). The peak fractions containing both proteins in the 1:1 ratio were collected and the concentration was estimated by $OD_{280}$ and the predicted extinction coefficient of 24,155 $M^{-1}cm^{-1}$. The sample was diluted to 40 nM in the GF buffer and a 20 μl aliquot was subjected to mass photometry immediately after the dilution (OneMP, Refeyn). For KRas4B[G12V] and Rgl2[RA]–Halo complex formation, both proteins were separately purified by SEC as described above and mixed at the 1:1 ratio to generate a premix 2 μM sample, which was further diluted to 20, 50, 100, and 250 nM before the measurement.

The measurement was carried out in conventional microscope cover glass (Marienfeld, no 1.5 H) cleaned by rinsing with deionized water (´5) and isopropanol (5) followed by drying under a $N_2$ flow, using a silicon gasket (Grace Bio-labs) to confine the sample. Adsorption of individual molecules of complex was detected across an imaging area of 10.8 μm by 2.9 mm.

Video recordings from interferometric scattering microscopy for a duration of 60 s were obtained and the single events corresponding to surface adsorption of the complex were identified using AcquireMP software (Refeyn). Data analysis was performed using DiscoverMP software (Refeyn) and OriginPro 2021 (OriginLab).

## Graphical representation of protein structures

Protein structure images were generated using PyMol (The PyMOL Molecular Graphics System, Version 1.2r3pre, Schrödinger, LLC). Electrostatic surface charge potential images were produced using PyMol vacuum electrostatics function. Amino acid residues in the interaction surfaces of protein complexes in a PDB format were predicted using LIGPLOT (Wallace et al, 1995).

# Data Availability

Structural statistics and coordinates for the KRAS4B$^{G12V}$–Rgl2$^{RA}$ crystal structure and the NMR Rgl2$^{RA}$ structure are available at PDB with the IDs 8B69 and 8AU4, respectively.

# Supplementary Information

# Acknowledgements

We thank the help and expert advice from the University of Leicester colleagues Peter Moody, Thomas Schalch, Ian Eperon, Mohammed Bhogadia, Luke Bailey, Kyle Daynes, Idir Malki, and Ziad Ibrahim. Biolayer interferometry experiments were conducted with the guidance of Holly Birchenough and Thomas Jowitt (Biomolecular Analysis Core Facility, University of Manchester). We gratefully acknowledge financial support from Wellcome Trust ISSF COVID career support and inclusion scheme 204801/Z/16/Z, University of Leicester MSc program (to A Hasan, MA Muro Campillo and K Tanaka), University of Leicester College PhD programme (to M Tariq, LR Abbott and K Tanaka), BBSRC MIBTP PhD programme BB/T00746X/1 (to S Mayooramurugan and K Tanaka), the Funding Programs for Core Research for Evolutional Science and Technology (CREST; JPMJCR13M3 to Y Ito, JPMJCR21E5 to T Ikeya) the Japan Science and Technology Agency (JST), Grants-in-Aid for Scientific Research (JP15K06979 to T Ikeya, JP19H05645 to Y Ito) and Scientific Research on Innovative Areas (JJP15H01645, JP16H00847, JP17H05887, and JP19H05773 to Y Ito P26102538, JP25120003, JP16H00779, and JP21K06114 to T Ikeya) from the Japan Society for the Promotion of Science (JSPS), Shimazu foundation, and the Precise Measurement Technology Promotion Foundation.

## Author Contributions

M Tariq: conceptualization and investigation.
T Ikeya, Y Ito, K Tanaka: conceptualization, investigation, and methodology.
N Togashi, S Kamei, S Mayooramurugan, LR Abbott, A Hasan, S Sukegawa, MA Muro Campillo: investigation.
L Fairall: validation, investigation, and methodology.
C Bueno-Alejo: investigation and methodology.
B Romartinez-Alonso, AJ Hudson, JWR Schwabe: methodology.
C Dominguez: conceptualization, validation, investigation, and methodology.

## Conflict of Interest Statement

The authors declare that they have no conflict of interest.

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
