## [Reviewer comments · Life Science Alliance]

Life Science Alliance

Structural insights into the complex of oncogenic K-Ras4BG12V and Rgl2, a RalA/B activator

Mishal Tariq, Teppei Ikeya, Naoyuki Togashi, Louise Fairall, Shun Kamei, Sannojah Mayooramurugan, Lauren Abbott, Anab Hasan, Carlos Bueno-Alejo, Sakura Sukegawa, Beatriz Romartinez-Alonso, Miguel Muro Campillo, Andrew Hudson, Yutaka Ito, John Schwabe, Cyril Dominguez, and Kayoko Tanaka

DOI: <https://doi.org/10.26508/lsa.202302080>

Corresponding author(s): *Kayoko Tanaka, University of Leicester*

Review Timeline:

Submission Date:	2023-04-06
Editorial Decision:	2023-05-12
Revision Received:	2023-08-08
Editorial Decision:	2023-09-06
Revision Received:	2023-09-21
Editorial Decision:	2023-09-22
Revision Received:	2023-09-28
Accepted:	2023-10-02

Scientific Editor: Novella Guidi

Transaction Report:

May 12, 2023

Re: Life Science Alliance manuscript #LSA-2023-02080

Dr. Kayoko Tanaka
University of Leicester
Department of Molecular and Cell Biology
Henry Wellcome Building
Lancaster Road
Leicester LE1 7HB
United Kingdom

Dear Dr. Tanaka,

Thank you for submitting your manuscript entitled "Structural insights into the complex of oncogenic K-Ras4BG12V and Rgl2, a RalA/B activator" to Life Science Alliance. The manuscript was assessed by expert reviewers, whose comments are appended to this letter. We invite you to submit a revised manuscript addressing the Reviewer comments.

Thank you for this interesting contribution to Life Science Alliance. We are looking forward to receiving your revised manuscript.

Sincerely,

B. MANUSCRIPT ORGANIZATION AND FORMATTING:

Reviewer #1 (Comments to the Authors (Required)):

The work in this manuscript aimed to evaluate the interaction between K-Ras and the Ral protein exchange factor, Rgl2. The authors discovered that a common oncogenic mutation (G12V) leads to a higher affinity of K-Ras for Rgl2, and they have solved a structure that suggests that the proteins form a tetramer. This is discussed appropriately in the context of the other structures that have been solved. This work would be of interest to the field, subject to some modifications.

The binding assays look convincing by BLI, and this is supported by the size exclusion chromatography showing that the G12V variant has a higher affinity than the WT K-Ras.

The presence of the tetramer in the structure was unexpected, although there was an earlier structure of Ras with RalGDS that had a similar tetramer in the crystal. To understand whether this is a general feature of some Ras binding proteins, it would be essential to know whether this is a true interaction, since this could be a crystal artefact. The authors do try to confirm the presence of the tetramer in solution, but some of the results are equivocal.

For the RGL2 titration data it is stated that (page 8 lines 7-10):

"Signals from the Rgl2RA residues in $\beta 1$ at the N-terminal end display the largest changes. This is consistent with our structures showing that the highly flexible N-terminal region of free Rgl2RA (Supplementary Fig. S4A) becomes rigid upon complex formation through its interaction with K-Ras4BG12V as observed in the crystal structure."

Looking at Fig 5C and S3A, do the changes in the Rgl2 N-terminus show that there is a RGL2-RGL2 interaction in solution? These residues are not involved in the Rgl2-K-Ras interface, so it provides more evidence for the proposed tetramer. Furthermore, the titration shows that the chemical shifts continue to change even after the stoichiometry is 1:1. Presumably this is because there is a second binding of a lower affinity. This should be discussed in the manuscript because it provides evidence for a more complicated stoichiometry than a simple 1:1. Isothermal titration calorimetry could provide a useful orthogonal assay for both the affinity and the stoichiometry of the interaction, as long as there are sufficient heat changes.

Why did the authors record HSQC experiments and not TROSY given the size of the complex? They may have been able to see more peaks in the Ras spectrum with a TROSY. Can the authors explain the difference between Ras, where the peaks disappeared and Rgl where they just got weaker?

Mass photometry shows evidence for the heterotetramer in solution. It is surprising that the authors cannot see the dimer at all - can they offer any explanation?

Minor:

Fig 3 is confusing: it has a panel at the top and then panels A and B, but all panels should be named (e.g. A, B, C).

Reviewer #2 (Comments to the Authors (Required)):

Review on the manuscript submitted by Tariq et al. entitled "Structural insights into the oncogenic..."

Ras is known for a long time as an oncogene responsible (together with other mutations) for the development of a variety of human tumors, overall involved in 30% of human cancer cases. The observations of the mode of complex formation between Ras and Rgl2 (potentially representing a whole class of Ras effectors) reported in this paper are most different from known Ras/effector complexes and yield new insights into the way Ras interacts with the huge variety of partners and into possibilities to interfere with inhibitors.

In their introduction the authors point out the prominent role of K-RasB in cellular signaling and tumor development. Amongst the various effector proteins of Ras they emphasize the activators (exchange factors) of RalA and RalB, like RalGDS and Rgl because hyperactivation of this pathway seems to be prevalent in tumorigenesis. It is pointed out that Rgl2 is one of the most important effectors and therefore the complex structure is mandatory to investigate.

The results of this study can be highlighted as follows:

- The complex structure shows the formation of a hetero tetramer (like Ras and RalGDS), 2 Ras and 2 Rgl2 molecules. This could turn out to be typical for RA type effectors in contrast to RBD type (like Raf). Most intriguingly, Ras (switch I and switch II) binds to both effector molecules.
- The affinity to Rgl2 is higher for the G12V mutant of KRas as compared to the wild type.
- The G12V position is buried in the Ras/Ras interface of the complex.

Critical points:

- The analysis of the BLI binding experiments (Fig1) is not clear:

- i) you talk about fast association and dissociation kinetics, but the association curve looks like a very slow process (usually this type of Ras/effector association takes place within milliseconds - depending on the conc.). In particular, an end of binding is not visible. Did you try to reach equilibrium or saturation? This brings me to the next point:
 - ii) plotting a response curve in order to obtain the Kd value makes only sense after reaching equilibrium. What exactly did you plot? A steady state rate? Or the response values after a given time? The term "steady-state analysis" in the text is not understandable.
 - iii) the dissociation phase appears a bit weird. Is it the sudden drop that corresponds to dissociation? Or is it the following, slow phase? Did you derive a rate constant from this? (Note: a fast diss. rate -like the fast drop- together with the very slow ass. process will not yield a Kd value close 1 μM)
 - iv) panel B right hand, bottom, why does the binding response start at such high levels (rather than 0)? Then, the drop is going much lower than the starting value, why? After the drop the response increases again (for the high conc.) How reproducible is all this?
 - v) Still referring to the same panel and the one above: how would the binding curve (upper panel) look like after subtracting the starting values each?
- It is surprising to see that despite a significant (but not a large) difference of Kd values for wt and G12V (6.1 μM vs 1.4 μM) there is such a (seemingly) clear cut difference in the elution behaviour of gel filtration (Fig S1), i.e. no complex in the case of wt. Was the nucleotide loading (GppNHp) checked? According to the elution profile together with the SDS gels there seems to be a large excess of Ras wt over Rgl2 while in the case of G12V the amounts of the two proteins seem to be similar.
- As to the question of a hetero dimer or tetramer complex in solution, the gel filtration experiments should give more precise conclusion on this. Did you calibrate the column with standard size proteins? Judging the elution profiles "by eye" it appears that a hetero dimer is formed rather than a tetramer.
- Along the same question of a tetramer in solution: can you give more detail in quantitative terms on the analysis of the "overall decrease in the NMR signals" or an analysis of line shapes in order to conclude as to dimer vs. tetramer.
- How do you envisage an increase in binding affinity to Rgl2 by the G12V mutant while this residue is located in the Ras/Ras interface?
- Finally, the binding curve presented in figure 1 does not show any sign of tetramer formation (like cooperativity). Which equation was used for the fit? It looks like one-site binding which does not reflect tetramer formation.

Minor:

On page 16: it is doubtful that the proteins were denatured by the addition of EDTA and renatured by MgCl₂ (or was this shown by CD or so?) Rather, EDTA captures Mg ions and in the absence of Mg ions the nucleotide affinity is reduced / dissociation rate increased...

Page 18 line 1: MgCl₂

Recommendations:

Amongst others there is two important findings in this work: i) 2 Ras/2 Rgl2 tetramer formation, ii) in Rgl2 binding the oncogene mutant G12V has higher affinity than wt

Ad i) While the gel filtration data do not support tetramer formation it is important that the other methods show this more convincingly. Improve the NMR analysis, make mass photometry more convincing by showing reference data / or mass calibration with standard proteins (even better well established protein complexes, e.g. Ras/Raf; arguing with dimer complex at the limit of detection is not convincing)

Ad ii) Think about a more suitable method to quantify affinity (like ITC) because your conclusions rely strongly on higher affinity of G12V.

Reviewer #3 (Comments to the Authors (Required)):

Tariq et al. present biochemical and structural data (both crystal- and NMR-structure) of the complex of K-RasG12V with the RA-domain of the RalA/B GEF Rgl2.

They show a 6-fold higher affinity of K-RasG12V for that domain than of the wt and can only crystallize the structure with the former, albeit K-Ras wt was mostly isolated in the GTP-state. The complex they identify is a dimer of a heterodimer, however, it is interpreted as a native heterotetramer. Contacts in this complex are then extensively described and the interaction is further validated by NMR-data. Finally, mass photometry experiments were performed that support the existence of a complex in the size range of a hetero-tetramer.

While the presentation of the work is good and ample figures are provided, this work lacks convincing experimental data and has two major conceptual flaws: First, why would an effector evolve to have highest affinity for an oncogenic version of K-Ras? Essentially all other such effector fragments (RBD- or RA-domains) bind with comparable affinity to all Ras-isoforms, consistent with the high sequence conservation in the switch I and II regions.

Second, the symmetrical hetero-tetramer is unlikely to be physiologically relevant given the arrangement of the domain in the complex, with the C-termini of the two K-Ras proteins that would anchor the protein to the membrane, pointing in opposite directions. In this regard, why would a dimer-interface have evolved to comprise the G12V-mutation?

Thus, data interpretation is implausible and the punch-line of the paper appears unlikely and artificially enhanced.

Major concerns:

-Additional binding data need to be provided that support the K-RasG12V-selectivity of the Rgl2-RA-domain. What is the affinity to other oncogenic mutants? What is the affinity to GDP-K-Ras? What is the affinity to other Ras-isoforms, is this really a K-Ras specific binder?

-All crucial interacting residues should be validated by mutational analysis for their effect on complex formation. Ideally this would be performed in the context of the full length Rgl2-protein, validating the claimed allele and isoform preferences.

-Complexation should be demonstrated with endogenous proteins where possible or at least in complex cell lysates.

-The binding curve in Fig.1B does not reach saturation. Additional binding data supporting their claim should be provided. They can include affinity measurements based on NMR-data.

-If the mass photometry detection starts only above 30 kDa, they would not be able to detect dimers. This could be fixed by examining complexes of the tagged constructs (GST-RA domain). These experiments should ideally be performed with the full-length proteins. Additional data supporting native heterotetramers are needed.

-Additional domains of the effector would almost certainly impact on the arrangement of the interaction partners, hence any of the major conclusion should be validated in the context of the full-length effector.

Minor comments:

- all gene names should be written in the conventional way, e.g. KRAS (not K-RAS)

- p.4 L7: unusual wording- 'ignite ERK...'

- instances of improper unit usage (nm instead of nM, e.g. p. 18 L4)

- the steady state affinity derivation from BLI-data is not described, even though these are the only quantitative biochemical data provided

We would like to thank all the reviewers for their valuable and constructive comments and suggestions. Specific responses to each comment are stated below.

Reviewer #1 (Comments to the Authors (Required)):

The work in this manuscript aimed to evaluate the interaction between K-Ras and the Ral protein exchange factor, Rgl2. The authors discovered that a common oncogenic mutation (G12V) leads to a higher affinity of K-Ras for Rgl2, and they have solved a structure that suggests that the proteins form a tetramer. This is discussed appropriately in the context of the other structures that have been solved. This work would be of interest to the field, subject to some modifications.

We thank the reviewer for his/her overall positive evaluation of our work.

The binding assays look convincing by BLI, and this is supported by the size exclusion chromatography showing that the G12V variant has a higher affinity than the WT K-Ras.

Thank you very much for the comment. However, in the revised version, we concluded that the overall affinity of the G12V and Rgl2 is comparable to the affinity of the KRas.WT and Rgl2. We improved the data quality during the revision process by conducting BLI using a new Octet R8 instrument. We also ensured that the WT KRas sample was successfully loaded with GMPPNP for the size exclusion chromatography (SEC) experiment. The revised data shows that the G12V mutation does alter the KRas-Rgl2 binding kinetics; both the k_{on} and k_{off} values increased in the presence of G12V (revised Fig. 1B). However, the overall K_D values of KRas-Rgl2 interaction for WT and G12V were comparable (revised supplementary Fig. S2), and the SEC profiles for WT and G12V were indistinguishable (revised supplementary Fig. S1).

The presence of the tetramer in the structure was unexpected, although there was an earlier structure of Ras with RalGDS that had a similar tetramer in the crystal. To understand whether this is a general feature of some Ras binding proteins, it would be essential to know whether this is a true interaction, since this could be a crystal artefact. The authors do try to confirm the presence of the tetramer in solution, but some of the results are equivocal.

For the RGL2 titration data it is stated that (page 8 lines 7-10):

"Signals from the Rgl2RA residues in β 1 at the N-terminal end display the largest changes. This is consistent with our structures showing that the highly flexible N-terminal region of free Rgl2RA

(Supplementary Fig. S4A) becomes rigid upon complex formation through its interaction with K-Ras4B^{G12V} as observed in the crystal structure."

Looking at Fig 5C and S3A, do the changes in the Rgl2 N-terminus show that there is a RGL2-RGL2 interaction in solution? These residues are not involved in the Rgl2-K-Ras interface, so it provides more evidence for the proposed tetramer. Furthermore, the titration shows that the chemical shifts continue to change even after the stoichiometry is 1:1. Presumably this is because there is a second binding of a lower affinity. This should be discussed in the manuscript because it provides evidence for a more complicated stoichiometry than a simple 1:1. Isothermal titration calorimetry could provide a useful orthogonal assay for both the affinity and the stoichiometry of the interaction, as long as there are sufficient heat changes.

Thank you for highlighting this interesting point. We believe that our original description of this point was somewhat unclear and did not capture the complete picture of the data. For this reason, we have now revised the manuscript to clarify what the Rgl2 N-terminus signal change implies, as below.

"Signals from the Rgl2^{RA} residues in β 1 at the N-terminal end (642-649) display the largest changes. This is in agreement with the KRas4B^{G12V}:Rgl2^{RA} heterotetramer crystal structure where the highly flexible N-terminal region of free Rgl2^{RA} (Supplementary Fig. S6A) becomes rigid through interaction with another Rgl2 molecule in the heterotetramer (Fig. 4B). In addition, the overall decrease in the NMR signals is compatible with the formation of a relatively large complex such as the KRas4B^{G12V}:Rgl2^{RA} heterotetramer complex (62kDa). Meanwhile, the titration experiment showed that the chemical shift changes were not fully saturated in most of the Rgl2^{RA} residues when the Rgl2^{RA}:Kras^{G12V} molar ratio was 1:1 or even 1:2 (Fig. 5C), suggesting that the stoichiometry is more complex than simple 1:1, possibly due to the heterotetramer formation."

We appreciate the valuable suggestion of the reviewer that we conduct ITC experiments. We attempted ITC experiments using various concentration combinations of KRas and Rgl2. Unfortunately, we could not obtain reliable data since the heat changes were relatively small. We show below an example of our best-looking attempts which could not provide reliable information.

Why did the authors record HSQC experiments and not TROSY given the size of the complex? They may have been able to see more peaks in the Ras spectrum with a TROSY.

Thank you very much for the suggestion to conduct the TROSY experiment. We attempted the TROSY of ^2H - ^{15}N - ^{13}C -KRas in the presence of Rgl2^{RBD}, as shown below. The signal quality was improved, but the number of invisible residues was still comparable to the HSQC spectra; hence the data could not facilitate the identification of the residues at the interface. In addition, as the yield of the labelled sample was low, we could not conduct rigorous titration experiments. Therefore, we did not explore this strategy further.

[Figure removed by editorial staff per authors' request]

Can the authors explain the difference between Ras, where the peaks disappeared and Rgl where they just got weaker?

The most prominent signal “disappearance” of Ras is seen when the GDP-bound and GMPPNP-bound Ras molecules are compared. Ras signal disappearance occurs when it is bound by GMPPNP (Please see the supplementary fig. S7).

With respect to the signal intensity fold-change/disappearance upon the complex formation, we believe that both Rgl2 and KRas show a similar trend, as shown in Figures 5 and 6.

Mass photometry shows evidence for the heterotetramer in solution. It is surprising that the authors cannot see the dimer at all - can they offer any explanation?

The lowest molecular weight the mass photometry system can detect is at about 30 kDa, which is the expected molecular weight of the 1:1 heterodimer complex of KRas.GV (~20 kDa) and Rgl2-RA domain (~10 kDa). We believe that this was the main reason we could not detect the dimer.

We repeated the mass photometry experiment using the KRas.GV (~20 kDa) and Halo-tagged Rgl2-RA (~46 kDa). At 50 nM, we detected the formation of complexes, the molecular weights of which are in the range of the heterodimer. At higher concentrations (100-250 nM), we could detect the formation of complexes of larger sizes that are in the range of the heterotetramer. The result is presented in the revised Fig. 7.

Minor:

Fig 3 is confusing: it has a panel at the top and then panels A and B, but all panels should be named (e.g. A, B, C).

Thank you very much for pointing this out. We have corrected the labelling.

Reviewer #2 (Comments to the Authors (Required)):

Review on the manuscript submitted by Tariq et al. entitled "Structural insights into the oncogenic..."

Ras is known for a long time as an oncogene responsible (together with other mutations) for the development of a variety of human tumors, overall involved in 30% of human cancer cases. The observations of the mode of complex formation between Ras and Rgl2 (potentially representing a

whole class of Ras effectors) reported in this paper are most different from known Ras/effector complexes and yield new insights into the way Ras interacts with the huge variety of partners and into possibilities to interfere with inhibitors.

In their introduction the authors point out the prominent role of K-RasB in cellular signaling and tumor development. Amongst the various effector proteins of Ras they emphasize the activators (exchange factors) of RalA and RalB, like RalGDS and Rgl because hyperactivation of this pathway seems to be prevalent in tumorigenesis. It is pointed out that Rgl2 is one of the most important effectors and therefore the complex structure is mandatory to investigate.

The results of this study can be highlighted as follows:

- The complex structure shows the formation of a hetero tetramer (like Ras and RalGDS), 2 Ras and 2 Rgl2 molecules. This could turn out to be typical for RA type effectors in contrast to RBD type (like Raf). Most intriguingly, Ras (switch I and switch II) binds to both effector molecules.
- The affinity to Rgl2 is higher for the G12V mutant of KRas as compared to the wild type.
- The G12V position is buried in the Ras/Ras interface of the complex.

Critical points:

- The analysis of the BLI binding experiments (Fig1) is not clear:
 - i) you talk about fast association and dissociation kinetics, but the association curve looks like a very slow process (usually this type of Ras/effector association takes place within milliseconds - depending on the conc.). In particular, an end of binding is not visible. Did you try to reach equilibrium or saturation? This brings me to the next point:

Thank you very much for raising this point. The BLI diagram (before the revision) showed the Association and Dissociation phases only, without the Baseline phase, as indicated in the below diagram. In the diagram, the data of all the concentrations were aligned at the end of the Baseline (time -5 to 0 (sec)). The data showed a very rapid signal increase at the start of the Association phase; hence we described the kon rate to be very fast. Because of this behaviour, the result looked as if the binding level was already high at time 0.

Original data

However, as you point out, both samples did not reach an endpoint; we initially ascribed that to some drift caused by the equipment rather than to the mode of binding. Our interpretation turned out to involve a misjudgement for the WT sample when we repeated the experiment (please see below for details).

Fortunately, we could access a newer BLI machine (Octet R8), and we repeated the experiment multiple times with an improved experimental condition. The result is presented in the revised Fig. 1.

ii) plotting a response curve in order to obtain the Kd value makes only sense after reaching equilibrium. What exactly did you plot? A steady state rate? Or the response values after a given time? The term "steady-state analysis" in the text is not understandable.

Thank you for flagging up this important point. As explained above, in the pre-revised figure, we misjudged that the end of the Association phase represented the endpoint. Under this misjudgement, we used the values obtained at the end of the Association phase (Time 295-299

(sec) in the figure above) for the “steady-state analysis”, where the binding signals and analyte concentrations were plotted to deduce the K_D value.

We repeated the experiment using an improved condition and a new Octet R8 instrument; we found that the WT sample did not reach the endpoint. Therefore, for the revised Fig. 1, we did not conduct the steady-state analysis.

iii) iii) the dissociation phase appears a bit weird. Is it the sudden drop that corresponds to dissociation? Or is it the following, slow phase? Did you derive a rate constant from this? (Note: a fast diss. rate -like the fast drop- together with the very slow ass. process will not yield a K_d value close 1 μM)

Thank you very much for highlighting this intriguing binding mode. After repeating the experiment multiple times using an improved condition and a new Octet R8 instrument, we concluded that the wildtype KRas interaction with Rgl2^{RA} involves something beyond homogenous 1:1 binding. The model fitting looks better when we apply the heterogenous 2:1 binding model or the 1:2 bivalent binding model (the RSS value improved from over 90 to less than 2, and the R-squared value improved from 0.9284 to over 0.99). However, these two models simulate completely different sample statuses, and therefore, at this stage, we are still open to various possibilities that cause the peculiar binding mode of KRas.WT and Rgl2^{RA}. The model fitting result is shown in the new Supplementary Figure S2A.

Meanwhile, the KRas.G12V x Rgl2^{RA} sample fits well with the 1:1 binding model, and the 2:1 heterogenous binding model provides only marginal improvement (Supplementary Fig. S2B).

We also addressed whether this binding issue is common to a prototype Ras-effector pair, KRas and BRAF^{RBD}. In this case, we observed a very similar mode of KRas-BRAF^{RBD} binding for both the wildtype and G12V mutant, indicating that the G12V-induced kinetics change is specific to the KRas-Rgl2 pair. The result is shown in the revised Fig. 1, and the model fitting analysis is shown in the revised Supplementary Fig. S3.

All together, the result supports the working hypothesis that the G12V mutation influences the binding kinetics between KRas4B and Rgl2^{RA}. The mechanism of how the G12V influences the binding kinetics still requires further studies.

iv) iv) panel B right hand, bottom, why does the binding response start at such high levels (rather

than 0)? Then, the drop is going much lower than the starting value, why? After the drop the response increases again (for the high conc.) How reproducible is all this?

Thank you very much for pointing out the issues associated with panel B, right hand, which represented the KRas.G12V sample. As mentioned in section i), the binding response in this diagram looked to have started at a high level, as the association occurred very rapidly.

We repeated the experiment multiple times using the newer BLI equipment (Octet R8). We believe that the raised issues were caused because of some instability of the old BLI equipment. A representative result produced by the new Octet R8 is provided in the revised Fig. 1.

v) Still referring to the same panel and the one above: how would the binding curve (upper panel) look like after subtracting the starting values each?

Thank you for the comment. The binding curve was already presented in a way where the starting values were subtracted from each sample of the varying concentrations.

However, as explained above, the data quality of the BLI result was improved using a new Octet R8 instrument and the revised result is presented in the revised Fig. 1.

-It is surprising to see that despite a significant (but not a large) difference of K_D values for wt and G12V (6.1 μM vs 1.4 μM) there is such a (seemingly) clear cut difference in the elution behaviour of gel filtration (Fig S1), i.e. no complex in the case of wt. Was the nucleotide loading (GppNHp) checked? According to the elution profile together with the SDS gels there seems to be a large excess of Ras wt over Rgl2 while in the case of G12V the amounts of the two proteins seem to be similar.

Thank you very much for the comment. We did not check the nucleotide-binding status of this experiment; we repeated the experiment using the KRas4B.WT and KRas4B.G12V samples where the GMPPNP loading was confirmed. We found that the SEC profiles of WT+Rgl2^{RA} and G12V+Rgl2^{RA} are very similar. The result is consistent with our revised BLI result, where the K_D values of these two cases are in the same range. Supplementary Fig. S1 is revised with the new data.

-As to the question of a hetero dimer or tetramer complex in solution, the gel filtration experiments

should give more precise conclusion on this. Did you calibrate the column with standard size proteins? Judging the elution profiles "by eye" it appears that a hetero dimer is formed rather than a tetramer.

Thank you very much for the comment. Yes, we confirmed that the main elution peak corresponds to a hetero dimer rather than a tetramer. Therefore, if a tetramer exists, its population is low or unstable. The fast k_{on} and k_{off} kinetics seen in the BLI experiment are consistent with this interpretation.

-Along the same question of a tetramer in solution: can you give more detail in quantitative terms on the analysis of the "overall decrease in the NMR signals" or an analysis of line shapes in order to conclude as to dimer vs. tetramer.

Thank you very much for raising this issue. In order to obtain more insights into the issues of "dimer vs. tetramer", we conducted the NMR relaxation experiment and modelling for the τ_c (rotational correlation time) of the G12V:Rgl2 complex, assuming that the dimer and tetramer conformations are roughly spherical. The estimated τ_c is closer to the size of the heterodimer structure. However, the approximate radii of gyration obtained here may not fully reflect the respective structures, since the structures of the heterodimer and heterotetramer in the crystal are not completely spherical and are close to ellipsoidal. Additionally, considering the possibility of an equilibrium-like exchange process between the heterodimer and heterotetramer, or transient tetramer formation, further NMR relaxation measurements and detailed analysis would be needed to obtain a more accurate estimation. The data is presented in the revised Supplementary Fig. S8.

-How do you envisage an increase in binding affinity to Rgl2 by the G12V mutant while this residue is located in the Ras/Ras interface?

Thank you very much for raising this important issue. A working hypothesis may be that the G12V mutation might affect the overall structure of KRas4B in solution, and therefore, the binding kinetics to Rgl2 may also be affected. Our result of the KRas4B^{WT} and KRas4B^{G12V} CD spectra, where improved structural stability was seen for KRas4B^{G12V} (Fig. 1 F), might be consistent with this prediction. However, further future studies are needed to fully understand the effect of the G12V mutation.

-Finally, the binding curve presented in figure 1 does not show any sign of tetramer formation (like cooperativity). Which equation was used for the fit? It looks like one-site binding which does not reflect tetramer formation.

We revised the BLI result. Intriguingly, the sensorgram of the KRas.WT x Rgl2^{RA} can be model-fitted by the 1:2 bivalent binding by the data analysis software (Octet Analysis Studio, ver.13.0, Sartorius), whereas the KRas.GV x Rgl2^{RA} is fitted by the 1:1 binding (Fig. S2A, the third panel). A bold hypothesis could be that the wild-type KRas might bind to Rgl2^{RA} through two distinct steps, but the G12V mutation could help bypass the first step. However, at this stage we can only speculate and we need to address this point in the future with new measurements.

Minor:

On page 16: it is doubtful that the proteins were denatured by the addition of EDTA and renatured by MgCl₂ (or was this shown by CD or so?) Rather, EDTA captures Mg ions and in the absence of Mg ions the nucleotide affinity is reduced / dissociation rate increased...

Thank you very much for the comment. We removed the term "denature".

Page 18 line 1: MgCl₂

Thank you very much for the comment. We corrected the typo.

Recommendations:

Amongst others there is two important findings in this work: i) 2 Ras/2 Rgl2 tetramer formation, ii) in Rgl2 binding the oncogene mutant G12V has higher affinity than wt

Ad i) While the gel filtration data do not support tetramer formation it is important that the other methods show this more convincingly. Improve the NMR analysis, make mass photometry more convincing by showing reference data / or mass calibration with standard proteins (even better well established protein complexes, e.g. Ras/Raf; arguing with dimer complex at the limit of detection is not convincing)

Ad ii) Think about a more suitable method to quantify affinity (like ITC) because your conclusions rely strongly on higher affinity of G12V.

Thank you very much for all the useful comments.

In the revised manuscript, we conducted additional experiments, as listed below, to address the concerns.

(1) ITC did not give conclusive data (as responded to reviewer #1).

(2) As mentioned earlier, we repeated the BLI experiment. The result showed that the G12V mutation alters the binding kinetics between KRas and Rgl2^{RA}. Importantly, the G12V did not cause such an effect on the KRas · BRAF^{RBD} interaction.

(3) We conducted an additional mass photometry experiment using a Halo-tagged Rgl2^{RA}, following Reviewer #3's helpful suggestion. Under this experimental condition, we observed the dimer and tetramer formation.

From these results, together with the crystallisation and NMR studies, we conclude that the KRas4B^{G12V}-Rgl2^{RA} complex may exist as multiple statuses, including a dimer and a tetramer, the proportion of which might be influenced by the G12V oncogenic mutation.

Reviewer #3 (Comments to the Authors (Required)):

Tariq et al. present biochemical and structural data (both crystal- and NMR-structure) of the complex of K-RasG12V with the RA-domain of the RalA/B GEF Rgl2.

They show a 6-fold higher affinity of K-RasG12V for that domain than of the wt and can only crystallize the structure with the former, albeit K-Ras wt was mostly isolated in the GTP-state. The complex they identify is a dimer of a heterodimer, however, it is interpreted as a native heterotetramer. Contacts in this complex are then extensively described and the interaction is further validated by NMR-data. Finally, mass photometry experiments were performed that support the existence of a complex in the size range of a hetero-tetramer.

While the presentation of the work is good and ample figures are provided, this work lacks convincing experimental data and has two major conceptual flaws: First, why would an effector evolve to have highest affinity for an oncogenic version of K-Ras? Essentially all other such effector fragments (RBD- or RA-domains) bind with comparable affinity to all Ras-isoforms, consistent with the high sequence conservation in the switch I and II regions.

Thank you very much for appreciating our data presentation and raising the important question of whether KRas-RBD/RA affinities change because of an oncogenic mutation. The issue was elegantly addressed by Smith and Ikura, who conducted "parallel NMR analyses" to reveal the

effector interaction hierarchies change upon introduction of the G12V mutation (Smith and Ikura, (2014) Nature Chemical Biology, 10, 223-230). We were inspired by their work and started to examine the interaction between KRas and Rgl2.

Second, the symmetrical hetero-tetramer is unlikely to be physiologically relevant given the arrangement of the domain in the complex, with the C-termini of the two K-Ras proteins that would anchor the protein to the membrane, pointing in opposite directions. In this regard, why would a dimer-interface have evolved to comprise the G12V-mutation?

Thus, data interpretation is implausible and the punch-line of the paper appears unlikely and artificially enhanced.

Thank you very much for the invaluable comments. We are also struck by the difference between the well-characterised Ras-RBD complexes and the KRas-Rgl2 complex, which is highly similar to the HRas-RALGDS complex. Intriguingly, Ikura and colleagues have reported that the membrane orientations of the KRas4B-ARAF^{FRBD} complex and the KRas4B-RALGDS^{RA} complex are distinct (Mazhab-Jafari et al., (2015) PNAS, 112, 6625–6630). These observations may indicate that the mode of Ras-effector interaction may vary among different effectors, and we believe that reporting our observation contributes to a better understanding of the issue. We would like to explore the in vivo status of KRas:Rgl2 in future, but it is beyond the scope of our present study.

Major concerns:

-Additional binding data need to be provided that support the K-RasG12V-selectivity of the Rgl2-RA-domain. What is the affinity to other oncogenic mutants? What is the affinity to GDP-K-Ras? What is the affinity to other Ras-isoforms, is this really a K-Ras specific binder?

Thank you very much for raising these points.

We confirmed that the GDP-bound KRas does not interact with Rgl2. This is shown in Supplementary Fig.S1.

In this manuscript, we do not claim that Rgl2 is specific to KRas.G12V. The KRAS4B is reported to represent more than half of all RAS transcripts (Newlaczyl et al., (2017) Sci Rep 7, 41297), and the G12V mutation is one of the most common KRAS mutations. Therefore, we focused on this variant for the interaction with Rgl2 as a model.

*Previously, mouse Rgl2 homologue, Rlf, was shown to interact with HRas, KRas and NRas (Esser et al., Biochemistry, **37**, 13453-13462, (1998), Bauer et al., JBC, **274**, 17763-17770 (1999), Ferro et al., Biochem J. (2008) **415**, 145-154). Therefore, we expect human Rgl2 to be able to bind other human Ras isoforms.*

-All crucial interacting residues should be validated by mutational analysis for their effect on complex formation. Ideally this would be performed in the context of the full length Rgl2-protein, validating the claimed allele and isoform preferences.

Thank you very much for the interesting suggestions.

We are not claiming any allele or isoform preference for Rgl2 in this report. So, mutating all the residues to examine the allele and isoform preferences is beyond the scope of our work.

-Complexation should be demonstrated with endogenous proteins where possible or at least in complex cell lysates.

Thank you very much for the comment. We agree that it is important to study the endogenous KRas-Rgl2 interaction. However, in this project, we examined the mode of binding between KRas and the RA of Rgl2 to obtain a basic biochemical property. The endogenous protein behaviour is beyond the scope of this project.

*Previously, in one of the BioID-based mass spectroscopy studies using bladder cancer cells, HRas.G12V was shown to interact with Rgl2 (Kovalski et al., (2019), Mol Cell, **73**, 830-844). Meanwhile, in the same study, KRas.G12D using colon cancer cells did not bring down Rgl2. Another BioID-based study using murine pancreatic ductal adenocarcinoma cells expressing KRAS4B.G12D did not detect Rgl2 (Chen et al., (2021) PNAS, **118**, e2016904118) and affinity-purification coupled with mass spectroscopy study of KRAS oncogenic mutants using human colon carcinoma cells did not identify Rgl2 (Catozzi et al., (2022) Cell Commun Signal 20: 24. doi:10.1186/s12964-022-00823-5, Ternet et al., (2023) Life Science Alliance, 6, e202201670). These observations indicate that the KRas-Rgl2 complex formation may be dependent on cell types and may be transient.*

-The binding curve in Fig.1B does not reach saturation. Additional binding data supporting their claim should be provided. They can include affinity measurements based on NMR-data.

Thank you very much for the comment. As mentioned in response to Reviewer #2, we improved the quality of the BLI data, which showed that the K_D values deduced from the k_{on} and k_{off} values are comparable between the KRas.WT and KRas.G12V. However, the k_{on} and k_{off} values were altered (increased) in the binding between the KRas.G12V mutant and Rgl2^{RA}. This effect was not seen in the complex between KRasWT/G12V and BRAF^{RBD}.

-If the mass photometry detection starts only above 30 kDa, they would not be able to detect dimers. This could be fixed by examining complexes of the tagged constructs (GST-RA domain). These experiments should ideally be performed with the full-length proteins. Additional data supporting native heterotetramers are needed.

Thank you very much for the helpful suggestion. We repeated the mass photometry experiment using a Halo-tagged Rgl2^{RA} construct. The result agrees with the formation of a dimer and a tetramer. We revised Fig. 7 with the updated result.

The purification of native heterotetramers is beyond the scope of this study.

-Additional domains of the effector would almost certainly impact on the arrangement of the interaction partners, hence any of the major conclusion should be validated in the context of the full-length effector.

Thank you very much for this constructive comment. We agree, and obviously we do not intend to overinterpret our data. We report that the kinetics of interaction between the RA domain of Rgl2 and KRas is altered when KRas carries the G12V mutation. Interestingly, our revised BLI result showed that the interaction between the BRAF^{RBD} and KRas is not affected by the G12V mutation. We believe that these pieces of information contribute to understanding the mechanism of Ras-mediated signalling and give further insight into the field.

Minor comments:

- all gene names should be written in the conventional way, e.g. KRAS (not K-RAS)

Thank you very much for pointing this out. We have revised the gene name to KRAS and the protein to KRas.

- p.4 L7: unususal wording- 'ignite ERK...'

We have revised it to “trigger ERK...”.

- instances of improper unit usage (nm instead of nM, e.g. p. 18 L4)

Thank you very much for pointing this out. We have revised the BLI data and used the appropriate units.

- the steady state affinity derivation from BLI-data is not described, even though these are the only quantitative biochemical data provided

Thank you very much for the comment. We have revised the BLI data and described in some detail how the data analysis was conducted.

September 6, 2023

Re: Life Science Alliance manuscript #LSA-2023-02080R

Dr. Kayoko Tanaka
University of Leicester
Department of Molecular and Cell Biology
Henry Wellcome Building
Lancaster Road
Leicester LE1 7HB
United Kingdom

Dear Dr. Tanaka,

Thank you for submitting your revised manuscript entitled "Structural insights into the complex of oncogenic K-Ras4BG12V and Rgl2, a RalA/B activator" to Life Science Alliance. The manuscript has been seen by the original reviewers whose comments are appended below. While the reviewers continue to be overall positive about the work in terms of its suitability for Life Science Alliance, some important issues remain.

Our general policy is that papers are considered through only one revision cycle; however, we are open to one additional short round of revision. Please note that I will expect to make a final decision without additional reviewer input upon re-submission.

Please submit the final revision within one month, along with a letter that includes a point by point response to the remaining reviewer comments.

To upload the revised version of your manuscript, please log in to your account: <https://lsa.msubmit.net/cgi-bin/main.plex>
You will be guided to complete the submission of your revised manuscript and to fill in all necessary information.

B. MANUSCRIPT ORGANIZATION AND FORMATTING:

Sincerely,

Reviewer #1 (Comments to the Authors (Required)):

Overall, the manuscript is much improved by the new experiments and the authors have addressed my comments. In particular,

the addition of the new mass photometry experiments has strengthened the evidence for a tetramer.

In light of the BLI data that shows that the WT and G12V mutants have similar affinities for Rgl1, the first section of the results should be retitled, since the current title:

'Active KRas4BG12V has a stronger affinity to Rgl2RA than KRas4BWT'.

Is misleading, because the affinities are the same.

I noted that the authors have now included some NMR relaxation data. The data were only recorded at one field, and were analysed using a simple isotropic, spheroid model that is not correct: as stated in the results, both the heterodimer and heterotetramer are ellipsoid. Furthermore, if both are present in solution and are interconverting, the analysis of these experiments is non-trivial. Given these limitations, I am not convinced that it is worth including the relaxation data as it does not add anything meaningful to the story.

Reviewer #2 (Comments to the Authors (Required)):

The authors added new experimental data which were analyzed and interpreted more carefully as compared to the previous version of the manuscript.

Reviewer #3 (Comments to the Authors (Required)):

The authors have provided a manuscript with minor improvements, which still leaves many questions unclear. The changes are not only difficult to follow without a marked manuscript with changes tracked, but also because the responses are not structured by numbering. Responses lack clarity, not saying clearly what has been changed where.

The manuscript remains mostly a big collection of structure representations in a well too high number of supplementary figures. The lack of decisive revision work is reflected in an abstract that essentially remained unaltered. The same pertains to the remainder of the manuscript, where just two new paragraphs are inserted that could also be shorter. It should be clear to the authors that if they for a certain reason do not consider it necessary to perform requested experiments, this should be reflected by an adjustment of the introduction, data presentation or discussion. Important plausibility concerns are just ignored.

Considering the structure is essentially identical to other Ras structures with Rgl1, the novelty of the current manuscript appears very limited.

I will in the following refer to the major concerns I raised previously, numbering them consecutively.

1- If extensive previous work exists of a Ras/ Rgl2 complexes/ interaction, then this certainly has to be introduced and it must be qualified, what the specific motivation for the current study is inside the manuscript. A response to the reviewer is insufficient.

2- A mutational analysis of the complex interface of Rgl2 (ideally full length) and KRasG12V, e.g. using IP should be a basic validation step. The authors have all the tools to perform such experiments.

3-The same for showing interactions of endogenous proteins, which cannot be evaded by stating 'beyond the scope'. If there is ample evidence for interaction, this should be introduced. If it is contentious, it is even more important to examine this here. If the work remains partially hypothetical, because no evidence for endogenous interaction of full length proteins was found, this has to be stated to the reader upfront and in the abstract.

I do not go as far as saying, why report the structure of a biologically not occurring complex, to preserve the authors efforts.

4- In Fig. 1B, new data are provided that are left in the raw data format, without compiling binding curves. Instead, a too large set of sensograms is provided in the supplementary without summarising clearly all the data provided.

5 and 6- Providing multiple data sets of the same truncated protein species (Rgl2-RA), is not helpful to understand what the oligomerization state of the native complex is. Without showing a tetramer with full-length Rgl2, I cannot accept their statement of heterotetramerization.

We thank all the reviewers for their valuable and constructive comments and suggestions. We are pleased to hear that, overall, the reviewers are positive with the revision we made. Below, we provide specific responses to each of the latest comments.

Reviewer #1 (Comments to the Authors (Required)):

Overall, the manuscript is much improved by the new experiments and the authors have addressed my comments. In particular, the addition of the new mass photometry experiments has strengthened the evidence for a tetramer.

In light of the BLI data that shows that the WT and G12V mutants have similar affinities for Rgl1, the first section of the results should be retitled, since the current title:

'Active KRas4BG12V has a stronger affinity to Rgl2RA than KRas4BWT'.

Is misleading, because the affinities are the same.

Thank you very much for pointing this out. We revised the subtitle to **“The G12V oncogenic mutation causes a change in the interaction kinetics between the KRas4B and the Rgl2^{RA}”**.

I noted that the authors have now included some NMR relaxation data. The data were only recorded at one field, and were analysed using a simple isotropic, spheroid model that is not correct: as stated in the results, both the heterodimer and heterotetramer are ellipsoid. Furthermore, if both are present in solution and are interconverting, the analysis of these experiments is non-trivial. Given these limitations, I am not convinced that it is worth including the relaxation data as it does not add anything meaningful to the story.

Thank you very much for clarifying the limitation of our NMR relaxation data. We agree with you about the limitation. However, we also believe that this approximate estimation clearly highlights the fact that, under the experimental condition of this NMR analysis, the status of the complex is not a stable heterotetramer, although further analyses are clearly needed for more accurate and robust conclusions. So, we would like to leave the data as supplementary information with a revised, simplified description in the main text. We also would like to note that the NMR relaxation experiment was originally suggested by Reviewer 2, who is happy with our revision.

Reviewer #2 (Comments to the Authors (Required)):

The authors added new experimental data which were analyzed and interpreted more carefully as compared to the previous version of the manuscript.

Thank you very much for the positive comments. We are pleased to hear that our revision has appropriately addressed the raised concerns.

Reviewer #3 (Comments to the Authors (Required)):

The authors have provided a manuscript with minor improvements, which still leaves many questions unclear. The changes are not only difficult to follow without a marked manuscript with changes tracked, but also because the responses are not structured by numbering. Responses lack clarity, not saying clearly what has been changed where.

The manuscript remains mostly a big collection of structure representations in a well too high number of supplementary figures.

The lack of decisive revision work is reflected in an abstract that essentially remained unaltered. The same pertains to the remainder of the manuscript, where just two new paragraphs are inserted that could also be shorter. It should be clear to the authors that if they for a certain reason do not consider it necessary to perform requested experiments, this should be reflected by an adjustment of the introduction, data presentation or discussion. Important plausibility concerns are just ignored.

We appreciate Reviewer 3's comments, and during the first revision round, we responded to each comment without ignoring them. We are grateful for the comments, as we have improved the quality of the manuscript substantially by following Reviewer 3's comments.

The central theme of this study has been to present the structural and biochemical information of the KRAS^{G12V} - Rgl2^{RA} complex, and this remains the same before and after the revision. Therefore, the abstract does not have substantial changes.

Following Reviewer 3's suggestion, we now provide an additional short paragraph in the Introduction about the *in vitro* and *in vivo* interaction between Ras and Rgl2 (and its mouse homologue, Rlf).

Considering the structure is essentially identical to other Ras structures with Rgl1, the novelty of the current manuscript appears very limited.

We appreciate the comment. However, we believe that providing the structural and biochemical properties of the KRas^{G12V} - Rgl2^{RA} complex substantially improves the current understanding of the mechanism of KRas signalling for the following reasons. First, together with the previously studied HRas-RALGDS^{RA} complex, our result demonstrates that the Ras-RalGEF^{RA} complex can form the unique tetramer structure where the oncogenic mutation valine12 resides at the Ras:Ras interface. This structural feature was not seen in the KRas-Rgl1^{RA} structures. Second, the G12V mutation alters the binding kinetics between KRas and Rgl2^{RA}. Such change does not occur in the KRas-BRAF^{RBD} complex (this study) or in the KRas-Rgl1^{RA} complex (Eves et al., 2022).

I will in the following refer to the major concerns I raised previously, numbering them consecutively.

1- If extensive previous work exists of a Ras/ Rgl2 complexes/ interaction, then this certainly has to be introduced and it must be qualified, what the specific motivation for the current study is inside the manuscript. A response to the reviewer is insufficient.

Thank you very much for clarifying this point. In the latest manuscript, an additional paragraph is provided in the Introduction summarising the previous studies on the *in vitro* and *in vivo* Ras/Rgl2(Rlf) complexes. We also explained that our research goal is to elucidate the interface of the Ras/Rgl2 complex at an atomic level and whether the oncogenic G12V mutation influences the interaction.

2- A mutational analysis of the complex interface of Rgl2 (ideally full length) and KRasG12V, e.g. using IP should be a basic validation step. The authors have all the tools to perform such experiments.

Thank you very much for the valuable suggestion. As the Kd value is relatively high (~1.5 μM), we found it technically challenging to conduct quantitative IP experiments reliably. Therefore, we plan to conduct a systematic mutational analysis of KRas and Rgl2 involving BLI, NMR and mass

photometry. However, such a set of experiments will be a substantial work, and we aim to generate a set of data to be published as the follow-up work.

3-The same for showing interactions of endogenous proteins, which cannot be evaded by stating 'beyond the scope'. If there is ample evidence for interaction, this should be introduced. If it is contentious, it is even more important to examine this here.

If the work remains partially hypothetical, because no evidence for endogenous interaction of full length proteins was found, this has to be stated to the reader upfront and in the abstract.

I do not go as far as saying, why report the structure of a biologically not occurring complex, to preserve the authors efforts.

Thank you very much for raising this important point. As stated above, we revised the Introduction and cited relevant studies that showed the endogenous *in vivo* interaction.

4- In Fig. 1B, new data are provided that are left in the raw data format, without compiling binding curves. Instead, a too large set of sensograms is provided in the supplementary without summarising clearly all the data provided.

Thank you very much for the comment. It is unclear what Reviewer 3 means by “compiling binding curves”, but if it would be “steady-state” analysis curves, we do not provide them as the wildtype case does not reach a steady state, so conducting the steady-state analysis is inappropriate. This issue was raised by Reviewer 2 in the previous revision round, which helped us not to run the steady-state analysis, but to conduct the *kon/koff* kinetics analysis carefully.

We included the K_D , *kon*, *koff* and RSS values in Fig. 1 and clearly stated what these values mean in the main text. We also combined previous supplementary Figures S2 and S3 to make the figures concise and clear.

5 and 6- Providing multiple data sets of the same truncated protein species (Rgl2-RA), is not helpful to understand what the oligomerization state of the native complex is. Without showing a tetramer with full-length Rgl2, I cannot accept their statement of heterotetramerization.

Thank you very much for the comment. We agree with Reviewer 3 that it is very important to look at the structural arrangement of the full-length Rgl2 in complex with KRas. However, we also believe that the structural arrangement of the KRas-Rgl2^{RA} complex is highly informative, as it

shows a striking similarity with the HRas-RALGDS^{RA} structure (PDB 1LFD), which had been an outlier among other Ras-effector complexes. In the follow-up study, we plan to analyse the full-length Rgl2 and KRas complex.

September 22, 2023

RE: Life Science Alliance Manuscript #LSA-2023-02080RR

Dr. Kayoko Tanaka
University of Leicester
Department of Molecular and Cell Biology
Henry Wellcome Building
Lancaster Road
Leicester LE1 7HB
United Kingdom

Dear Dr. Tanaka,

Thank you for submitting your revised manuscript entitled "Structural insights into the complex of oncogenic K-Ras4BG12V and Rgl2, a RalA/B activator". We would be happy to publish your paper in Life Science Alliance pending final revisions necessary to meet our formatting guidelines.

- please add the Twitter handle of your host institute/organization as well as your own or/and one of the authors in our system
- please upload your Tables in editable .doc or excel format;
- there is a name discrepancy for one of your co-authors. In the system, it's Shun Kamei, but in the manuscript file, it's Syun Kamei. Please correct this.
- please consult our manuscript preparation guidelines <https://www.life-science-alliance.org/manuscript-prep> and make sure your manuscript sections are in the correct order
- please incorporate any points from the Conclusion section into the Discussion; we only allow a Discussion section
- please add your main, supplementary figure, and table legends to the main manuscript text after the references section
- we encourage you to revise the figure legends for figures 1, and 5 such that the figure panels are introduced in an alphabetical order
- please add callouts for Figures S4A-C, S14A-B to your main manuscript text
- the 3 References listed separately should be incorporated into the main reference list

A. FINAL FILES:

B. MANUSCRIPT ORGANIZATION AND FORMATTING:

Sincerely,

October 2, 2023

RE: Life Science Alliance Manuscript #LSA-2023-02080RRR

Dr. Kayoko Tanaka
University of Leicester
Department of Molecular and Cell Biology
Henry Wellcome Building
Lancaster Road
Leicester LE1 7HB
United Kingdom

Dear Dr. Tanaka,

Thank you for submitting your Research Article entitled "Structural insights into the complex of oncogenic K-Ras4BG12V and Rgl2, a RalA/B activator". It is a pleasure to let you know that your manuscript is now accepted for publication in Life Science Alliance. Congratulations on this interesting work.

DISTRIBUTION OF MATERIALS:

Again, congratulations on a very nice paper. I hope you found the review process to be constructive and are pleased with how the manuscript was handled editorially. We look forward to future exciting submissions from your lab.

Sincerely,
